# Meta Two-Sample Testing:
# Learning Kernels for Testing with Limited Data

**Feng Liu**[*]
Australian AI Institute, UTS
`feng.liu@uts.edu.au`

**Wenkai Xu**[*]
Gatsby Unit, UCL[†]
`xwk4813@gmail.com`

**Jie Lu**
Australian AI Institute, UTS
`jie.lu@uts.edu.au`

**Danica J. Sutherland**
UBC and Amii[‡]
`dsuth@cs.ubc.ca`

## Abstract

Modern kernel-based two-sample tests have shown great success in distinguishing complex, high-dimensional distributions by learning appropriate kernels (or, as a special case, classifiers). Previous work, however, has assumed that many samples are observed from both of the distributions being distinguished. In realistic scenarios with very limited numbers of data samples, it can be challenging to identify a kernel powerful enough to distinguish complex distributions. We address this issue by introducing the problem of *meta two-sample testing (M2ST)*, which aims to exploit (abundant) auxiliary data on related tasks to find an algorithm that can quickly identify a powerful test on new target tasks. We propose two specific algorithms for this task: a generic scheme which improves over baselines, and a more tailored approach which performs even better. We provide both theoretical justification and empirical evidence that our proposed meta-testing schemes outperform learning kernel-based tests directly from scarce observations, and identify when such schemes will be successful.

## 1 Introduction

Two-sample tests ask, "given samples from each, are these two populations the same?" For instance, one might wish to know whether a treatment and control group differ. With very low-dimensional data and/or strong parametric assumptions, methods such as $t$-tests or Kolmogorov-Smirnov tests are widespread. Recent work in statistics and machine learning has sought tests that cover situations not well-handled by these classic methods [1–18], providing tools useful in machine learning for domain adaptation, causal discovery, generative modeling, fairness, adversarial learning, and more [19–34]. Perhaps the most powerful known widely-applicable scheme is based on a kernel method known as the *maximum mean discrepancy* (MMD) [1] – or, equivalently [35], the energy distance [3] – when one *learns* an appropriate kernel for the task at hand [10, 16]. Here, one divides the observed data into "training" and "testing" splits, identifies a kernel on the training data by maximizing a power criterion $\hat{J}$, then runs an MMD test on the testing data (as illustrated in Figure 1a). This method generally works very well when enough data is available for both training and testing.

In real-world scenarios, however, two-sample testing tasks can be challenging if we do not have very many data observations. For example, in medical imaging, we might face two small datasets of lung

---

[*]These authors contributed equally.

[†]Now at Department of Statistics, University of Oxford.

[‡]Work done partially while at the Toyota Technological Institute at Chicago.

35th Conference on Neural Information Processing Systems (NeurIPS 2021).

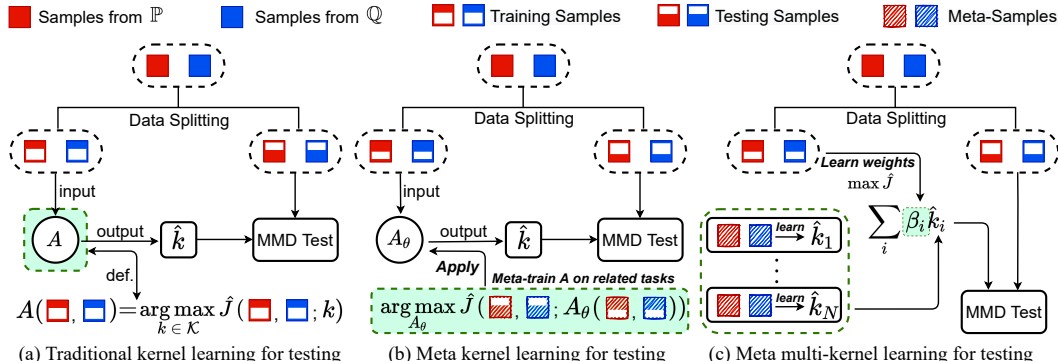

Figure 1: Comparison among (a) traditional kernel learning [10, 16], (b) meta kernel learning, and (c) meta multi-kernel learning for kernel two-sample testing, where $\hat{k}$ or $\hat{k}_i$ are the learned kernel.

computed tomography (CT) scans of patients with coronavirus diseases, and wish to know if these patients are affected in different ways. If they are from different distributions, the virus causing the disease may have mutated. Here, previous tests are likely to be relatively ineffective; we cannot learn a powerful kernel to distinguish such complex distributions with only a few observations.

In this paper, we address this issue by considering a problem setting where related testing tasks are available. We use those related tasks to identify a kernel selection *algorithm*. Specifically, instead of using a fixed algorithm $A$ to learn a kernel ("maximize $\hat{J}$ among this class of deep kernels"), we want to learn an algorithm $A_\theta$ from auxiliary data (Figure 1b):

$$\arg\max_{A_\theta} \mathbb{E}_{(\mathbb{P},\mathbb{Q})\sim\tau}\left[\hat{J}(S_\mathbb{P}^{te}, S_\mathbb{Q}^{te}; A_\theta(S_\mathbb{P}^{tr}, S_\mathbb{Q}^{tr}))\right]. \tag{1}$$

Here $\mathbb{P}, \mathbb{Q}$ are distributions sampled from a meta-distribution of related tasks $\tau$, which asks us to distinguish $\mathbb{P}$ from $\mathbb{Q}$. The corresponding observed sample sets $S_\mathbb{P}, S_\mathbb{Q}$ are split into training ($tr$) and testing ($te$) components. $A_\theta$ is a kernel selection algorithm which, given the two training sets (also called "support sets" in meta-learning parlance), returns a kernel function. $\hat{J}$ finally estimates the power of that kernel using the test set ("query sets"). In analogy with meta-learning [36–42], we call this learning procedure *meta kernel learning* (Meta-KL). We can then apply $A_\theta$ to select a kernel on our actual testing task, then finally run an MMD test as before (Figure 1b).

The adaptation performed by $A_\theta$, however, might still be very difficult to achieve with few training observations; even the best $A_\theta$ found by a generic adaptation scheme might over-fit to $S_\mathbb{P}^{tr}, S_\mathbb{Q}^{tr}$. For more stable procedures and, in our experiments, more powerful final tests, we propose *meta multi-kernel learning* (Meta-MKL). This algorithm independently finds the most powerful kernel for each related task; at adaptation time, we select a convex combination of those kernels for testing (Figure 1c), as in standard multiple kernel learning [43] and similarly to ensemble methods in few-shot classification [44, 45]. Because we are only learning a small number of weights rather than all of the parameters of a deep network, this adaptation can quickly find a high-quality kernel.

We provide both theoretical and empirical evidence that Meta-MKL is better than generic Meta-KL, and that both outperform approaches that do not use related task data, in low-data regimes. We find that learned algorithms can output kernels with high test power using only a few samples, where "plain" kernel learning techniques entirely fail.

## 2 Preliminaries

We will now review the setting of learning a kernel for a two-sample test, following [16]. Let $\mathcal{X} \subset \mathbb{R}^d$ and $\mathbb{P}, \mathbb{Q}$ be (unknown) Borel probability measures on $\mathcal{X}$, with $S_\mathbb{P} = \{\boldsymbol{x}_i\}_{i=1}^m \sim \mathbb{P}^m$ and $S_\mathbb{Q} = \{\boldsymbol{y}_j\}_{j=1}^m \sim \mathbb{Q}^m$ observed i.i.d. samples from these distributions. We operate in a classical hypothesis testing setup, with the null hypothesis that $\mathbb{P} = \mathbb{Q}$.

**Maximum mean discrepancy (MMD).** The basic tool we use is a kernel-based distance metric between distributions called the MMD, defined as follows. (The energy distance [3] is a special case of the MMD for a particular choice of $k$ [35].)

**Definition 1** (MMD [1]). *Let* $k : \mathcal{X} \times \mathcal{X} \to \mathbb{R}$ *be the bounded*[1] *kernel of an RKHS* $\mathcal{H}_k$ *(i.e.,* $\sup_{x,y \in \mathcal{X}} |k(x,y)| < \infty$*). Letting* $X, X' \sim \mathbb{P}$ *and* $Y, Y' \sim \mathbb{Q}$ *be independent random variables,*

$$\mathrm{MMD}(\mathbb{P}, \mathbb{Q}; k) = \sup_{f \in \mathcal{H}_k, \|f\|_{\mathcal{H}_k} \leq 1} \mathbb{E}[f(X)] - \mathbb{E}[f(Y)] = \sqrt{\mathbb{E}\left[k(X, X') + k(Y, Y') - 2k(X, Y)\right]}.$$

*If* $k$ *is* characteristic*, we have that* $\mathrm{MMD}(\mathbb{P}, \mathbb{Q}; k) = 0$ *if and only if* $\mathbb{P} = \mathbb{Q}$.

We can estimate MMD using the following $U$-statistic estimator, which is unbiased for $\mathrm{MMD}^2$ (denoted by $\widehat{\mathrm{MMD}}_u^2$) and has nearly minimal variance among unbiased estimators [1]:

$$\widehat{\mathrm{MMD}}_u^2(S_{\mathbb{P}}, S_{\mathbb{Q}}; k) := \frac{1}{m(m-1)} \sum_{i \neq j} H_{ij}^{(k)}, \tag{2}$$

$$H_{ij}^{(k)} := k(\boldsymbol{x}_i, \boldsymbol{x}_j) + k(\boldsymbol{y}_i, \boldsymbol{y}_j) - k(\boldsymbol{x}_i, \boldsymbol{y}_j) - k(\boldsymbol{y}_i, \boldsymbol{x}_j). \tag{3}$$

**Testing.** Under the null hypothesis $\mathfrak{H}_0$, $m\widehat{\mathrm{MMD}}_u^2$ converges in distribution as $m \to \infty$ to some distribution depending on $\mathbb{P}$ and $k$ [1, Theorem 12]. We can thus build a test with $p$-value equal to the quantile of our test statistic $m\widehat{\mathrm{MMD}}_u^2$ under this distribution. Although there are several methods to estimate this null distribution, it is usually considered best [10] to use a permutation test [46, 47]: under $\mathfrak{H}_0$, samples from $\mathbb{P}$ and $\mathbb{Q}$ are interchangeable, and repeatedly re-computing the statistic with samples randomly shuffled between $S_{\mathbb{P}}$ and $S_{\mathbb{Q}}$ estimates its null distribution.

**Test power.** We generally want to find tests likely to reject $\mathfrak{H}_0$ when indeed it holds that $\mathbb{P} \neq \mathbb{Q}$; the probability of doing so (for a particular $\mathbb{P}$, $\mathbb{Q}$, $k$ and $m$) is called *power*. For reasonably large $m$, Sutherland et al. [10] and Liu et al. [16] argue that the power is an almost-monotonic function of

$$J(\mathbb{P}, \mathbb{Q}; k) := \frac{\mathrm{MMD}^2(\mathbb{P}, \mathbb{Q}; k)}{\sigma_{\mathfrak{H}_1}(\mathbb{P}, \mathbb{Q}; k)}, \quad \sigma_{\mathfrak{H}_1}^2(\mathbb{P}, \mathbb{Q}; k) := 4\left(\mathbb{E}\left[H_{ij}^{(k)} H_{i\ell}^{(k)}\right] - \mathbb{E}\left[H_{ij}^{(k)}\right]^2\right). \tag{4}$$

Here, $\sigma_{\mathfrak{H}_1}^2$ is the asymptotic variance of $\sqrt{m}\,\widehat{\mathrm{MMD}}_u^2$ under $\mathfrak{H}_1$; it is defined in terms of an expectation of (3) with respect to the data samples $S_{\mathbb{P}}, S_{\mathbb{Q}}$, for $i, j, \ell$ distinct. The criterion (4) depends on the unknown distributions; we can estimate it from samples with the regularized estimator [16]

$$\hat{J}_\lambda(S_{\mathbb{P}}, S_{\mathbb{Q}}; k) := \widehat{\mathrm{MMD}}_u^2(S_{\mathbb{P}}, S_{\mathbb{Q}}; k) \Big/ \sqrt{\hat{\sigma}_{\mathfrak{H}_1}^2(S_{\mathbb{P}}, S_{\mathbb{Q}}; k) + \lambda}, \tag{5}$$

$$\hat{\sigma}_{\mathfrak{H}_1}^2(S_{\mathbb{P}}, S_{\mathbb{Q}}; k) := \frac{4}{m^3} \sum_{i=1}^{m}\left(\sum_{j=1}^{m} H_{ij}^{(k)}\right)^2 - \frac{4}{m^4}\left(\sum_{i=1}^{m}\sum_{j=1}^{m} H_{ij}^{(k)}\right)^2. \tag{6}$$

**Kernel choice.** Given two samples $S_{\mathbb{P}}$ and $S_{\mathbb{Q}}$, the best kernel is (essentially) the one that maximizes $J$ in (4). If we pick a kernel to maximize our estimate $\hat{J}$ using the same data that we use for testing, though, we will "overfit," and reject $\mathfrak{H}_0$ far too often. Instead, we use data splitting [2, 5, 10]: we partition the samples into two disjoint sets, $S_{\mathbb{P}} = S_{\mathbb{P}}^{tr} \cup S_{\mathbb{P}}^{te}$, obtain $k^{tr} = A(S_{\mathbb{P}}^{tr}, S_{\mathbb{Q}}^{tr}) \approx \arg\max_k \hat{J}_\lambda(S_{\mathbb{P}}^{tr}, S_{\mathbb{Q}}^{tr}; k)$, then conduct a permutation test based on $\mathrm{MMD}(S_{\mathbb{P}}^{te}, S_{\mathbb{Q}}^{te}; k^{tr})$. This process is summarized in Algorithm 2 and illustrated in Figure 1a.

This procedure has been successfully used not only to, e.g., pick the best bandwidth for a simple Gaussian kernel, but even to learn all the parameters of a kernel like (8) which incorporates a deep network architecture [10, 16]. As argued by Liu et al. [16], classifier two-sample tests [8, 48] (which test based on the accuracy of a classifier distinguishing $\mathbb{P}$ from $\mathbb{Q}$) are also essentially a special case of this framework – and more-general deep kernel MMD tests tend to work better. Although presented here specifically for $\widehat{\mathrm{MMD}}_u$, an analogous procedure has been used for many other problems, including other estimates of the MMD and closely-related quantities [2, 5, 49].

When data splitting, the training split must be big enough to identify a good kernel; with too few training samples $m^{tr}$, $\hat{J}_\lambda$ will be a poor estimator, and the kernel will overfit. The testing split,

---

[1]For the given expressions to exist and agree, we in fact only need Bochner integrability; this is implied by boundedness of either the kernel or the distribution, but can also hold more generally.

however, must also be big enough: for a given $\mathbb{P}$, $\mathbb{Q}$, and $k$, it becomes much easier to be confident that $\mathrm{MMD}(\mathbb{P}, \mathbb{Q}; k) > 0$ as $m^{te}$ grows and the variance in $\widehat{\mathrm{MMD}}_u(\mathbb{P}, \mathbb{Q}; k)$ accordingly decreases. When the number of available samples is small, both steps suffer. This work seeks methods where, by using related testing tasks, we can identify a good kernel with an extremely small $m^{tr}$; thus we can reserve most of the available samples for testing, and overall achieve a more powerful test.

Another class of techniques for kernel selection avoiding the need for data splitting is based on selective inference [15]. At least as studied by Kübler et al. [15], however, it is currently available only for restricted classes of kernels and with far-less-accurate "streaming" estimates of the MMD, which for fixed kernels can yield *far* less powerful tests than $\widehat{\mathrm{MMD}}_u$ [50]. In Section 5.4, we will demonstrate that in our settings, the data-splitting approach is empirically much more powerful.

## 3 Meta Two-Sample Testing

To handle cases with low numbers of available data samples, we consider a problem setting where *related* testing tasks are available. We use those tasks in a framework inspired by meta-learning [e.g. 36], where we use those related tasks to identify a kernel selection *algorithm*, as in (1). Specifically, we define a *task* as a pair $\mathcal{T} = (\mathbb{P}, \mathbb{Q})$ of distributions over $\mathcal{X}$ we would like to distinguish, and assume a meta-distribution $\tau$ over the space of tasks $\mathcal{T}$.

**Definition 2** (M2ST). *Assume we are assigned (unobserved) training tasks $\boldsymbol{T} = \{\mathcal{T}^i = (\mathbb{P}^i, \mathbb{Q}^i)\}_{i=1}^N$ drawn from a task distribution $\tau$, and observe meta-samples $\boldsymbol{S} = \{(S_{\mathbb{P}}^i, S_{\mathbb{Q}}^i)\}_{i=1}^N$ with $S_{\mathbb{P}}^i \sim (\mathbb{P}^i)^{n_i}$ and $S_{\mathbb{Q}}^i \sim (\mathbb{Q}^i)^{n_i}$. Our goal is to use these meta-samples to find a kernel learning algorithm $A_\theta$, such that for a target task $(\mathbb{P}', \mathbb{Q}') \sim \tau$ with samples $S_{\mathbb{P}'} \sim (\mathbb{P}')^{n'}$ and $S_{\mathbb{Q}'} \sim (\mathbb{Q}')^{n'}$, the learning algorithm returns a kernel $A_\theta(S_{\mathbb{P}'}, S_{\mathbb{Q}'})$ which will achieve high test power on $(\mathbb{P}', \mathbb{Q}')$.*

We measure the performance of $A_\theta$ based on the expected test power criterion for a target task:

$$\mathcal{J}(A_\theta, \tau) = \mathbb{E}_{(\mathbb{P}, \mathbb{Q}) \sim \tau}\left[\mathbb{E}_{S_{\mathbb{P}}^{tr} \sim \mathbb{P}^m, S_{\mathbb{Q}}^{tr} \sim \mathbb{Q}^m}\left[J(\mathbb{P}, \mathbb{Q}; A_\theta(S_{\mathbb{P}}^{tr}, S_{\mathbb{Q}}^{tr}))\right]\right]. \tag{7}$$

If $\tau$ were in some sense "uniform over all conceivable tasks," then a no-free-lunch property would cause M2ST to be hopeless. Instead, our assumption is that tasks from $\tau$ are "related" enough that we can make progress at improving (7).

By assuming the existence of a meta-distribution $\tau$ over tasks, it is promising to learn a general rule across different tasks [36]. Furthermore, we can quickly adapt to a solution to a specific task based on the learned rule [36], which is the reason why researchers have been focusing on meta-learning for several years. Specifically, in the meta two-sample testing, we hope to find "what differences between distributions generally look like" on the meta-tasks, and then at test time, use our very limited data to search for differences in that more constrained set of options. Because the space of candidate rules is more limited, we can (hopefully) find a good rule with a much smaller number of data points. For example, if we have meta-tasks which (like the target task) are determined only by a difference in means, then we want to learn a general rule that distinguishes between two samples by means. For a new task (i.e., new two samples), we hope to identify dimensions where two samples have different means, using only a few data points.

We will propose two approaches to finding an $A_\theta$. Neither is specific to any particular kernel parameterization, but for the sake of concreteness, we follow Liu et al. [16] in choosing the form

$$k_\omega(x, y) = [(1 - \epsilon)\kappa(\phi(x), \phi(y)) + \epsilon]\, q(x, y), \tag{8}$$

where $\phi$ is a deep neural network which extracts features from the samples, and $\kappa$ is a simple kernel (e.g., a Gaussian) on those features, while $q$ is a simple characteristic kernel (e.g. Gaussian) on the input space; $\epsilon \in (0, 1]$ ensures that every kernel of the form $k_\omega$ is characteristic. Here, $\omega$ represents all parameters in the kernel:[2] most parameters come from deep neural network $\phi$, but $\kappa$ and $q$ may have a few more parameters (e.g. length scales), and we can also learn $\epsilon$.

**Meta-KL.** We first propose Algorithm 1 as a standard approach to optimizing (7), à la MAML [36]: $A_\theta$ takes a small, fixed number of gradient ascent steps in $\hat{J}_\lambda$ for the parameters of $k_\omega$, starting

---

[2]We use $k_\omega$ when discussing issues relating to the parameters $\omega$. When no ambiguity arises, we use $k$ and $k_\omega$ interchangeably for deep neural network parameterized kernels. If we write only $k$, then $\omega$ still means the parameters of $k$ by default.

---

**Algorithm 1** Meta Kernel Learning (Meta-KL)

---

**Input:** Meta-samples $\boldsymbol{S} = \{(S_{\mathbb{P}i}, S_{\mathbb{Q}i})\}_{i=1}^{N}$; kernel architecture (8) and parameters $\omega_0$; regularization $\lambda$

**1. Initialize** algorithm parameters: $\theta := [\omega_{start}] \leftarrow [\omega_0]$

**2. Define** a parameterized learning algorithm $A_\theta(S_\mathbb{P}, S_\mathbb{Q})$ as:

   $\omega \leftarrow \omega_{start}$; **for** $t = 1, \ldots, n_{steps}$ **do** $\omega \leftarrow \omega + \eta \nabla_\omega \hat{J}_\lambda(S_\mathbb{P}, S_\mathbb{Q}; k_\omega)$; **end for**; **return** $k_\omega$

**for** $T = 1, 2, \ldots, T_{max}$ **do**

   **3: Sample** $\mathcal{I}$ as a set of indices in $\{1, 2, \ldots, N\}$ of size $n_{batch}$

   **for** $i \in \mathcal{I}$ **do**

      **4: Split** data as $S_{\mathbb{P}i} = S_{\mathbb{P}i}^{tr} \cup S_{\mathbb{P}i}^{te}$ and $S_{\mathbb{Q}i} = S_{\mathbb{Q}i}^{tr} \cup S_{\mathbb{Q}i}^{te}$;

      **5: Apply** the learning algorithm: $k_i \leftarrow A_\theta(S_{\mathbb{P}i}^{tr}, S_{\mathbb{Q}i}^{tr})$

   **end for**

   **6: Update** $\theta \leftarrow \theta + \beta \nabla_\theta \sum_{i \in \mathcal{I}} \hat{J}_\lambda(S_{\mathbb{P}i}^{te}, S_{\mathbb{Q}i}^{te}; k_i)$;           *# update $\theta$, i.e. start, to maximize $\mathcal{J}(A_\theta, \tau)$*

**end for**

**7: return** $A_\theta$

---

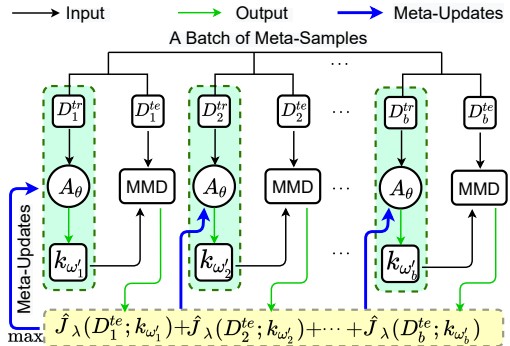

→ Input     → Output     → Meta-Updates

A Batch of Meta-Samples

$\max \hat{J}_\lambda(D_1^{te}; k_{\omega_1'}) + \hat{J}_\lambda(D_2^{te}; k_{\omega_2'}) + \cdots + \hat{J}_\lambda(D_b^{te}; k_{\omega_b'})$

Figure 2: Illustration of Algorithm 1.

---

**Algorithm 2** Testing with a Kernel Learner

---

**1: Input:** Two samples: $S_\mathbb{P}, S_\mathbb{Q}$; algorithm $A_\theta$

**2: Split** data as $S_\mathbb{P} = S_\mathbb{P}^{tr} \cup S_\mathbb{P}^{te}$ and $S_\mathbb{Q} = S_\mathbb{Q}^{tr} \cup S_\mathbb{Q}^{te}$;

**3: Learn** a kernel $k = A_\theta(S_\mathbb{P}^{tr}, S_\mathbb{Q}^{tr})$;

**4: Compute** $est \leftarrow \widehat{\mathrm{MMD}}_u^2(S_\mathbb{P}^{te}, S_\mathbb{Q}^{te}; k)$;

**for** $i = 1, 2, \ldots, n_{perm}$ **do**

   **5: Shuffle** $S_\mathbb{P}^{te} \cup S_\mathbb{Q}^{te}$ into $X$ and $Y$

   **6: Compute** $perm_i \leftarrow \widehat{\mathrm{MMD}}_u^2(X, Y; k)$;

**end for**

**7: Output:** p-value $\frac{1}{n_{perm}} \sum_{i=1}^{n_{perm}} \mathbb{1}(perm_i \geq est)$;

---

---

**Algorithm 3** Meta Multi-Kernel Learning (Meta-MKL)

---

**Input:** Meta-samples $\boldsymbol{S} = \{(S_{\mathbb{P}i}, S_{\mathbb{Q}i})\}_{i=1}^{N}$; kernel architecture as in (8)

**for** $i = 1, 2, \ldots, N$ **do**

   **1: Optimize** $\omega_i \leftarrow \widehat{\arg\max}_\omega \hat{J}_\lambda(S_{\mathbb{P}i}, S_{\mathbb{Q}i}; k_i)$ with some approximate optimization algorithm;

**end for**

**2: Define** $\mathcal{K} := \{\sum_{i=1}^{N} \beta_i k_i : \beta \in \mathbb{R}_{\geq 0}^{N}, \sum_i \beta_i = 1\}$;        *# convex combinations of kernels*

**3: return** the algorithm $A(S_\mathbb{P}, S_\mathbb{Q}) = \widehat{\arg\max}_{k \in \mathcal{K}} \hat{J}_\lambda(S_\mathbb{P}, S_\mathbb{Q}; k)$;

---

from a learned initialization point $\omega_{start} \in \theta$ (lines 1-2). We differentiate through $A_\theta$, and perform stochastic gradient ascent to find a good value of $\theta$ based on the meta-training sets (lines 3-6, also illustrated in Figure 2). Once we have learned a kernel selection procedure, we can again apply it to a testing task with Algorithm 2.

As we will see in the experiments, this approach does indeed use the meta-tasks to improve performance on target tasks. Differently from usual meta-learning settings, as in e.g. classification [36], however, here it is conceivable that there is a single good kernel that works for all tasks from $\tau$; improving on this single baseline kernel, rather than simply overfitting to the very few target points, may be quite difficult. Thus, in practice, the amount of adaptation that $A_\theta$ actually performs in its gradient ascent can be somewhat limited.

**Meta-MKL.** As an alternative approach, we also consider a different strategy for $A_\theta$ which may be able to adapt with many fewer data samples, albeit in a possibly weaker class of candidate kernels. Here, to select an $A_\theta$, we simply find the best kernel independently for each of the meta-training tasks. Then $A_\theta$ chooses the best *convex combination* of these kernels, as in classical multiple kernel learning [43] and similarly to ensemble methods in few-shot classification [45]. At adaptation time, we only attempt to learn $N$ weights, rather than adapting all of the parameters of a deep network; but, if the meta-training tasks contained some similar tasks to the target task, then we should be able to find a powerful test. This procedure is detailed in Algorithm 3 and illustrated in Figure 1c.

## 4 Theoretical analysis

We now analyze and compare the theoretical performance of direct optimizing the regularized test power from small sample size with our proposed meta-training procedures. To study our learning objective of approximate test power, we first state the following relevant technical assumptions, [16].

(A) The kernels $k_\omega$ are uniformly bounded as follows. For the kernels we use in practice, $\nu = 1$.

$$\sup_{\omega \in \Omega} \sup_{x \in \mathcal{X}} k_\omega(x, x) \le \nu.$$

(B) The possible kernel parameters $\omega$ lie in a Banach space of dimension $D$. Furthermore, the set of possible kernel parameters $\Omega$ is bounded by $R_\Omega$: $\Omega \subseteq \{\omega \mid \|\omega\| \le R_\Omega\}$.

(C) The kernel parameterization is Lipschitz: for all $x, y \in \mathcal{X}$ and $\omega, \omega' \in \Omega$,

$$|k_\omega(x, y) - k_{\omega'}(x, y)| \le L_k \|\omega - \omega'\|.$$

See Proposition 9 of Liu et al. [16] for bounds on these constants when using e.g. kernels of the form (8), in terms of the network architecture.

We will use $\sigma_\omega^2$ to refer to $\sigma_{\mathfrak{H}_1}^2(\mathbb{P}, \mathbb{Q}; k_\omega)$ of (4), and analogously for $\hat{\sigma}_\omega^2$, for the sake of brevity.

**Proposition 3** (Direct training with approximate test power, Theorem 6 of [16]). *Under Assumptions (A) to (C), suppose $\nu \ge 1$ is constant, and let $\tilde{\Omega}_s \subseteq \Omega$ be the set of $\omega$ for which $\sigma_\omega^2 \ge s^2$. Take the regularized estimate $\hat{\sigma}_{\omega,\lambda}^2 = \hat{\sigma}_\omega^2 + \lambda$ with $\lambda = m^{-1/3}$. Then, with probability at least $1 - \delta$,*

$$\sup_{\omega \in \tilde{\Omega}_s} \left| \hat{J}_\lambda(S_\mathbb{P}, S_\mathbb{Q}; k_\omega) - J(\mathbb{P}, \mathbb{Q}; k_\omega) \right| \le \xi_m = \mathcal{O}\left( \frac{1}{s^2 m^{1/3}} \left[ \frac{1}{s} + \sqrt{D \log(R_\Omega m) + \log \frac{1}{\delta}} + L_k \right] \right).$$

*Then, letting $\hat{k} \in \arg\max \hat{J}_\lambda(S_\mathbb{P}, S_\mathbb{Q}; \hat{k})$, we have $0 \le \sup_{k \in \mathcal{K}} J(\mathbb{P}, \mathbb{Q}; k) - J(\mathbb{P}, \mathbb{Q}; \hat{k}) \le 2\xi_m$.*

Since $m$ is small in our settings, and $s$ may also be small for deep kernel classes as noted by Liu et al. [16], this bound may not give satisfying results.

The key mechanism that drives meta-testing to work, intuitively, is training kernels on *related* tasks. How do we quantify the relatedness between different testing tasks?

**Definition 4** ($\gamma$-relatedness). *Let $(\mathbb{P}, \mathbb{Q})$ and $(\mathbb{P}', \mathbb{Q}')$ be the underlying distributions for two different two-sample testing tasks. We say the two tasks are $\gamma$-related w.r.t. learning objective $J$ if*

$$\sup_{k \in \mathcal{K}} |J(\mathbb{P}, \mathbb{Q}; k) - J(\mathbb{P}', \mathbb{Q}'; k)| = \gamma. \tag{9}$$

The relatedness measure is a (strong) assumption that two tasks are similar, because all kernels perform similarly on the two tasks, in terms of the approximate test power objective. It also implies the two problems are of similar difficulty, since for small $\gamma$, the ability of our MMD test statistics to distinguish the distributions (with optimal kernels) are similar.

**Definition 5** (Adaptation with Meta-MKL). *Given a set of kernels $\{k_1, \dots, k_N\}$, the Meta-MKL adaptation is the kernel $\hat{k} = \sum_{i=1}^N \hat{\beta}_i k_i$, where $\hat{\beta} = \arg\max_\beta \hat{J}_{\lambda^{ne}}(S_\mathbb{P}^{tr}, S_\mathbb{Q}^{tr}; \sum_i \beta_i k_i)$.*

This adaptation step uses the same learning objective, $\hat{J}_\lambda$, as directly training a deep kernel in Proposition 3 (though with a potentially different regularization parameter, $\lambda^{ne}$).

To analyze the Meta-MKL scheme, we will make the following assumption, which Proposition 26 in Liu et al. [16] shows implies Assumptions (A) to (C) with $\nu = KR_B\sqrt{N}$ and $L_k = K\sqrt{N}$.

(D) Let $\{k_i\}_{i=1}^N$ be a set of base kernels, each satisfying $\sup_{x \in \mathcal{X}} k_i(x, x) \le K$ for some finite constant $K$. Define the parameterized kernel $k$ as

$$k_\beta(x, y) = \sum_{i=1}^N \beta_i k_i(x, y) \tag{10}$$

where $\beta \in \mathbb{R}^N$, and let $B$ be the set of parameters $\beta$ such that $k_\beta$ is positive semi-definite (guaranteed if each $\beta_i \ge 0$) and $\|\beta\| \le R_B$ for some $R_B < \infty$.

**Theorem 6** (Performance of Meta-MKL). *Suppose we have $N$ meta-training tasks $\{(\mathbb{P}^i, \mathbb{Q}^i)\}_{i \in [N]}$, with corresponding optimal kernels $k_i^* \in \arg\max_{k \in \mathcal{K}} J(\mathbb{P}^i, \mathbb{Q}^i; k)$, and use $n$ samples to learn kernels $\hat{k}_i \in \arg\max_{k \in \mathcal{K}} \hat{J}_\lambda(S_{\mathbb{P}^i}, S_{\mathbb{Q}^i}; k)$ in the setting of Proposition 3. Let $(\mathbb{P}, \mathbb{Q})$ be a test task, with optimal kernel $k^* \in \arg\max_{k \in \mathcal{K}} J(\mathbb{P}, \mathbb{Q}; k)$, from which we observe $m$ samples $S_{\mathbb{P}}, S_{\mathbb{Q}}$. Call the Meta-MKL adapted kernel $\hat{k}_{\hat{\beta}} = \sum_i \hat{\beta}_i \hat{k}_i$, as in (10), with $\hat{\beta}$ found subject to Assumption (D). Let $(\mathbb{P}^j, \mathbb{Q}^j)$ be a meta-task which is $\gamma$-related to $(\mathbb{P}, \mathbb{Q})$. Then, with probability at least $1 - 2\delta$,*

$$J(\mathbb{P}, \mathbb{Q}; k^*) - J(\mathbb{P}, \mathbb{Q}; \hat{k}_\beta) \leq 2(\gamma + \xi_n^j + \xi_m)$$

*where $\xi_n^j$ is the bound of Proposition 3 for learning a kernel on $(\mathbb{P}^j, \mathbb{Q}^j)$, and $\xi_m$ is the equivalent bound for multiple kernel learning on $(\mathbb{P}, \mathbb{Q})$, which has $D = N$, $R_\Omega = R_B$, and $L_k = K\sqrt{N}$.*

The $\xi_n^j$ term depends on the meta-training sample size $n \gg m$. With enough (relevant) meta-training tasks (as $N$ grows), $\gamma$ is expected to go to $0$. So, the overall uniform convergence bound is likely to be dominated by the term $\xi_m$, giving an overall $m^{-1/3}$ rate: the same as Proposition 3 obtains for directly training a deep kernel only on $(\mathbb{P}, \mathbb{Q})$. This is roughly to be expected; similar optimization objectives are applied for both learning and adaptation, which are limited by sample size $m$. However, the other components of $\xi_m$ are likely much smaller than the corresponding parts of $\xi_n^j$ where the kernels are defined by a deep network: the variance must be lower-bounded over a much larger set of kernels, $D$ will be the number of parameters in the network rather than the number of meta-tasks, and the bound on $L_k$ from Proposition 23 of Liu et al. [16] is exponential in the depth of the network. Altogether, we expect MKL adaptation to be much more efficient than direct training.

We also expect that Theorem 6 is actually quite loose. The proof (in Appendix A) decomposes the loss relative to $\hat{k}_j$, picking just a single related kernel; it does not attempt to analyze how combining multiple kernels can improve the test power, because doing so in general seems difficult. Given this limitation, however, we also prove in Appendix A a bound on the adaptation scheme which explicitly only picks the single best kernel from the meta-tasks (Theorem 10), which is of a similar form to Theorem 6 but with the $\xi_m$ term replaced with one even better.

## 5 Experiments

Following Liu et al. [16], we compare the following baseline tests with our methods: 1) **MMD-D**: MMD with a deep kernel whose parameters are optimized; 2) **MMD-O**: MMD with a Gaussian kernel whose lengthscale is optimized; 3) Mean embedding (**ME**) test [5, 51]; 4) Smooth characteristic functions (**SCF**) test [5, 51], and 5) Classifier two-sample tests, including **C2ST-S** [8] and **C2ST-L** as described in Liu et al. [16]. None of these methods use related tasks at all, so we additionally consider an aggregated kernel learning (**AGT-KL**) method, which optimizes a deep kernel of the form (8) by maximizing the value of $\hat{J}_\lambda$ averaged over all the related tasks in the meta-training set $\mathbf{S}$.

For synthetic datasets, we take a single sample set for $S_{\mathbb{P}}^{tr}$ and $S_{\mathbb{Q}}^{tr}$, and learn parameters once for each method on that training set. We then evaluate its rejection rate (power or Type-I error, depending on if $\mathbb{P} = \mathbb{Q}$) using 100 new sample sets $S_{\mathbb{P}}^{te}, S_{\mathbb{Q}}^{te}$. For real datasets, we train on a subset of the available data, then evaluate on 100 random subsets, disjoint from the training set, of the remaining data. We repeat this full process 20 times for synthetic datasets or 10 times for real datasets, and report the mean rejection rate of each test and the standard error of the mean rejection rate. Implementation details are in Appendix B.1; the code is available at `github.com/fengliu90/MetaTesting`.

### 5.1 Results on Synthetic Data

We use a bimodal Gaussian mixture dataset proposed by [16], known as *high-dimensional Gaussian mixtures* (HDGM): $\mathbb{P}$ and $\mathbb{Q}$ subtly differ in the covariance of a single dimension pair. Here we consider only $d = 2$, since with very few samples the problem is already extremely difficult. Specifically,

$$\mathbb{P} = \frac{1}{2}\mathcal{N}\left(\begin{bmatrix} 0 \\ 0 \end{bmatrix}, \begin{bmatrix} 1 & 0 \\ 0 & 1 \end{bmatrix}\right) + \frac{1}{2}\mathcal{N}\left(\begin{bmatrix} 0.5 \\ 0.5 \end{bmatrix}, \begin{bmatrix} 1 & 0 \\ 0 & 1 \end{bmatrix}\right),$$

$$\mathbb{Q}(\Delta^h) = \frac{1}{2}\mathcal{N}\left(\begin{bmatrix} 0 \\ 0 \end{bmatrix}, \begin{bmatrix} 1 & -\Delta^h \\ -\Delta^h & 1 \end{bmatrix}\right) + \frac{1}{2}\mathcal{N}\left(\begin{bmatrix} 0.5 \\ 0.5 \end{bmatrix}, \begin{bmatrix} 1 & \Delta^h \\ \Delta^h & 1 \end{bmatrix}\right).$$

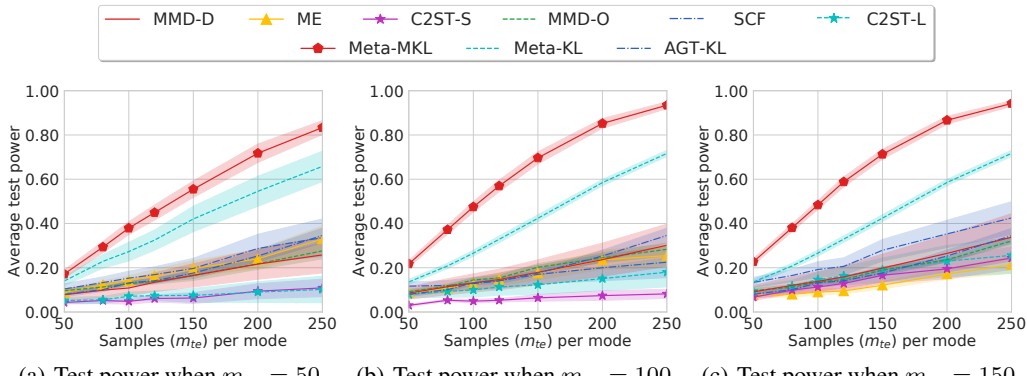

(a) Test power when $m_{tr} = 50$.   (b) Test power when $m_{tr} = 100$.   (c) Test power when $m_{tr} = 150$.

Figure 3: Test power on synthetic datasets for $\alpha = 0.05$. Average test power when increasing the number of testing samples $m_{te}$ while only using the 50 (a), 100 (b) and 150 (c) training samples per mode (i.e., $m_{tr} = 50, 100, 150$). Shaded regions show standard errors for the mean.

In this paper, our target task is $\mathcal{T} = (\mathbb{P}, \mathbb{Q}(0.7))$ and meta-samples are drawn from the $N = 100$ meta tasks $\boldsymbol{T} = \{\mathcal{T}^i := (\mathbb{P}, \mathbb{Q}(0.3 + 0.1 \times i/N))\}_{i=1}^N$; note that the target task is well outside the scope of training tasks. To evaluate all tests given limited data, we set the number of training samples $(S_\mathbb{P}^{tr}, S_\mathbb{Q}^{tr})$ to 50, 100, 150 per mode, and the number of testing samples $(S_\mathbb{P}^{te}, S_\mathbb{Q}^{te})$ from 50 to 250.

Figure 3 illustrates test powers of all tests. Meta-MKL and Meta-KL are the clear winners, with both tests much better when $m_{te}$ is over 100 per mode. It is clear that previous kernel-learning based tests perform poorly due to limited training samples. Comparing Meta-MKL with Meta-KL, apparently, we can obtain much higher power when we consider using multiple trained kernels. Although AGT-KL performs better than baselines, it cannot adapt to the target task very well: it only cares about "in-task" samples, rather than learning to adapt to new distributions. In Appendix B.2, we report the test power of our tests when increasing the number of tasks $N$ from 20 to 150. The results show that that increasing the number of meta tasks will help improve the test power on the target task.

## 5.2   Distinguishing CIFAR-10 or -100 from CIFAR-10.1

We distinguish the standard datasets of CIFAR-10 and CIFAR-100 [52] from the attempted replication CIFAR-10.1 [53], similar to Liu et al. [16]. Because only a relatively small number of CIFAR-10.1 samples are available, it is of interest to see whether by meta-training only on CIFAR-10's training set (as described in Appendix B), we can find a good test to distinguish CIFAR-10.1, with $m^{tr} \in \{100, 200\}$. Testing samples (i.e., $S_\mathbb{P}^{te}$ and $S_\mathbb{Q}^{te}$) are from test sets of each dataset. We report test powers of all tests with 200, 500, 900 testing samples in Table 1 (CIFAR-10 compared to CIFAR-10.1) and Table 2 (CIFAR-100 compared to CIFAR-10.1). Since Liu et al. [16] have shown that CIFAR-10 and CIFAR-10.1 come from different distributions, higher test power is better in both tables. The results demonstrate that our methods have much higher test power than baselines, which is strong evidence that leveraging samples from related tasks can boost test power significantly. Interestingly, C2ST tests almost entirely fail in this setting (as also seen by Recht et al. [53, Appendix B.2.8]); it is hard to learn useful information with only a few data points. In Appendix B.3, we also report results when meta-samples are generated by the training set of CIFAR-100 dataset.

## 5.3   Analysis of Closeness between Meta Training and Testing

This subsection studies how closeness between related tasks and the target task affects test powers of our tests. Given the target task $\mathcal{T}$ in synthetic datasets, we define tasks $\boldsymbol{T}$ with closeness $C$ as

$$\boldsymbol{T}(C) = \{\mathcal{T}^i := (\mathbb{P}, \mathbb{Q}((0.6 - C) + 0.1 \times i/N))\}_{i=1}^N. \tag{11}$$

It is clear that $\boldsymbol{T}(0)$ will contain our target task $\mathcal{T}$ (i.e., the closeness is zero). We also estimate the $\gamma$-relatedness between the target task and $\boldsymbol{T}(C)$, where $C \in \{0.1, 0.2, 0.3, 0.4, 0.5\}$, and the results show that $\gamma$ grows roughly linearly with $C$. Specifically, for $C = 0.1, 0.2, 0.3, 0.4, 0.5$, the estimate $\hat{\gamma}$ is $0.035, 0.067, 0.076, 0.104, 0.134$, respectively. (Details can be found in Appendix B.4.)

Table 1: Test power of tests on CIFAR-10 vs CIFAR-10.1 given very limited training data ($\alpha = 0.05$, $m_{tr} = 100, 200$). The $m_{te}$ represents number of samples when testing. Bold represents the highest mean per column.

| Methods | $m_{tr} = 100$ | | | $m_{tr} = 200$ | | |
|---|---|---|---|---|---|---|
| | $m_{te} = 200$ | $m_{te} = 500$ | $m_{te} = 900$ | $m_{te} = 200$ | $m_{te} = 500$ | $m_{te} = 900$ |
| ME | 0.084±0.009 | 0.096±0.016 | 0.160±0.035 | 0.104±0.013 | 0.202±0.020 | 0.326±0.039 |
| SCF | 0.047±0.013 | 0.037±0.011 | 0.047±0.015 | 0.026±0.009 | 0.018±0.006 | 0.026±0.012 |
| C2ST-S | 0.059±0.009 | 0.062±0.007 | 0.059±0.007 | 0.052±0.011 | 0.054±0.011 | 0.057±0.008 |
| C2ST-L | 0.064±0.009 | 0.064±0.006 | 0.063±0.007 | 0.075±0.014 | 0.066±0.011 | 0.067±0.008 |
| MMD-O | 0.091±0.011 | 0.141±0.009 | 0.279±0.018 | 0.084±0.007 | 0.160±0.011 | 0.319±0.020 |
| MMD-D | 0.104±0.007 | 0.222±0.020 | 0.418±0.046 | 0.117±0.013 | 0.226±0.021 | 0.444±0.037 |
| AGT-KL | 0.170±0.032 | 0.457±0.052 | 0.765±0.045 | 0.152±0.023 | 0.463±0.060 | 0.778±0.050 |
| Meta-KL | 0.245±0.010 | 0.671±0.026 | 0.959±0.013 | 0.226±0.015 | 0.668±0.032 | 0.972±0.006 |
| Meta-MKL | **0.277±0.016** | **0.728±0.020** | **0.973±0.008** | **0.255±0.020** | **0.724±0.026** | **0.993±0.003** |

Table 2: Test power of tests on CIFAR-100 vs CIFAR-10.1 given very limited training data ($\alpha = 0.05$, $m_{tr} = 100, 200$). The $m_{te}$ represents number of samples when testing. Bold represents the highest mean per column.

| Methods | $m_{tr} = 100$ | | | $m_{tr} = 200$ | | |
|---|---|---|---|---|---|---|
| | $m_{te} = 200$ | $m_{te} = 500$ | $m_{te} = 900$ | $m_{te} = 200$ | $m_{te} = 500$ | $m_{te} = 900$ |
| ME | 0.211±0.020 | 0.459±0.045 | 0.751±0.054 | 0.236±0.033 | 0.512±0.076 | 0.744±0.090 |
| SCF | 0.076±0.027 | 0.132±0.050 | 0.240±0.095 | 0.136±0.036 | 0.245±0.066 | 0.416±0.114 |
| C2ST-S | 0.064±0.007 | 0.063±0.010 | 0.067±0.008 | 0.324±0.034 | 0.237±0.030 | 0.215±0.023 |
| C2ST-L | 0.089±0.010 | 0.077±0.010 | 0.075±0.010 | 0.378±0.042 | 0.273±0.032 | 0.262±0.023 |
| MMD-O | 0.214±0.012 | 0.624±0.013 | 0.970±0.005 | 0.199±0.016 | 0.614±0.017 | 0.965±0.006 |
| MMD-D | 0.244±0.011 | 0.644±0.030 | 0.970±0.010 | 0.223±0.016 | 0.627±0.031 | 0.975±0.006 |
| AGT-KL | 0.596±0.044 | 0.979±0.010 | **1.000±0.000** | 0.635±0.038 | **0.994±0.002** | **1.000±0.000** |
| Meta-KL | 0.771±0.018 | 0.999±0.001 | **1.000±0.000** | 0.806±0.017 | **1.000±0.000** | **1.000±0.000** |
| Meta-MKL | **0.820±0.015** | **1.000±0.000** | **1.000±0.000** | **0.838±0.017** | **1.000±0.000** | **1.000±0.000** |

(a) Closeness is 0.1.   (b) Closeness is 0.2.   (c) Closeness is 0.3.

Figure 4: Test power when changing closeness between related tasks and the target task ($\alpha = 0.05$). Average test power when closeness is $0.1$ (a), $0.2$ (b) and $0.3$ (c), where only using the $50$ training samples per mode (i.e., $m_{tr} = 50$). Shaded regions show standard errors for the mean.

In Figure 4, we illustrate the test power of our tests when setting closeness $C$ to 0.1, 0.2 and 0.3, respectively. It can be seen that Meta-MKL and Meta-KL outperforms AGT-KL all the figures, meaning that Meta-MKL and Meta-KL actually learn algorithms that can quickly adapt to new tasks. Another phenomenon is that the gap between test powers of meta based KL and AGT-KL will get smaller if the closeness is smaller, which is expected since AGT-KL has seen closer related tasks.

## 5.4 Ablation Study

In previous sections, we mainly compare with previous kernel-learning tests and have shown that the test power can be improved significantly by our proposed tests. We now show that each component in our tests is effective to improve the test power.

First, we show that MKL can help improve the test power, and the data splitting used in Meta-MKL is much better than using the recent test of Kübler et al. [15]. The comparison has been made in synthetic datasets studied in Section 5.1 and the results can be found in Table 3. Meta-MKL-A is a test that takes all $\beta_i = 1/N$ in Algorithm 3, so that kernels are weighted equally. AGT-MKL uses

Table 3: The test power of various tests on the synthetic dataset, where for data-splitting methods we use only 10 samples to select kernels and 200 to test. Meta-MKL-A, AGT-MKL-A and Meta-MKL$_{SI}$ use 210 test samples. Bold represents the highest mean.

| Tests | Meta-MKL | Meta-MKL-A | Meta-KL | AGT-MKL | AGT-MKL-A | AGT-KL | Meta-MKL$_{SI}$ |
|---|---|---|---|---|---|---|---|
| Power | **0.792±0.014** | 0.780±0.012 | 0.509±0.046 | 0.364±0.016 | 0.358±0.021 | 0.253±0.025 | 0.058±0.008 |

Table 4: The test power of various tests on the CIFAR-10 vs CIFAR-10.1 task ($m_{tr} = 100$). Bold represents the highest mean per row.

| Tests | Meta-MKL | Meta-KL | AGT-KL | MMD-D w/ AC | MMD-D | MMD-O |
|---|---|---|---|---|---|---|
| $m_{te} = 200$ | **0.277±0.016** | 0.245±0.010 | 0.170±0.032 | 0.134±0.010 | 0.104±0.007 | 0.091±0.011 |
| $m_{te} = 500$ | **0.728±0.020** | 0.671±0.026 | 0.457±0.052 | 0.325±0.028 | 0.222±0.020 | 0.141±0.009 |
| $m_{te} = 900$ | **0.973±0.008** | 0.959±0.013 | 0.765±0.045 | 0.745±0.049 | 0.418±0.046 | 0.279±0.018 |

multiple kernels in AGT-KL (learning weights like Meta-MKL), and AGT-MKL-A does not learn the weights but just assigns weights $1/N$ directly to all base kernels. Meta-MKL$_{SI}$ is a kernel two-sample test using the selective inference technique of Kübler et al. [15] rather than data splitting in its $A_\theta$.

From Table 3, we can see that introducing the multiple kernel learning (MKL) scheme substantially improves test power, as it combines useful features learned from base kernels covering different aspects of the problems. Moreover, learning with approximate test power with data-splitting in the meta-setting also outperforms the non-splitting testing procedure MetaMKL$_{SI}$, since MetaMKL$_{SI}$ requires a linear estimator of MMD. The result also indicates that leveraging related tasks is also important to improve the test power, even though we only need a small set of training samples.

Then, we show that the labels used for constructing meta-samples are useful in the CIFAR datasets. We consider another test here: *MMD-D with all CIFAR-10* (MMD-D w/ AC), which runs the MMD-D test using the same sample from CIFAR-10 as did the meta-learning over all tasks together. Compared to Meta-MKL, Meta-KL and AGT-KL, MMD-D w/ AC does not use the label information contained in the CIFAR-10 dataset. The test power of MMD-D w/ AC is shown in Table 4. We can see that the test power of MMD-D w/ AC clearly outperforms MMD-D/MMD-O since MMD-D w/ AC sees more CIFAR-10 data in the training process. It is also clear that our methods still perform much better than MMD-D w/ AC. This result shows that the improvement of our tests does not solely come from seeing more data from CIFAR-10. Instead, the assigned labels for the meta-tasks indeed help.

# 6 Conclusions

This paper proposes kernel-based non-parametric testing procedures to tackle practical two-sample problems where the sample size is small. By meta-training on related tasks, our work opens a new paradigm of applying learning-to-learn schemes for testing problems, and the potential of very accurate tests in some small-data regimes using our proposed algorithms.

It is worth noting, however, that statistical tests are perhaps particularly ripe for mis-application, e.g. by over-interpreting small marginal differences between sample populations of people to claim "inherent" differences between large groups. Future work focusing on reliable notions of interpretability in these types of tests is critical. Meta-testing procedures, although they yield much better tests in our domains, may also introduce issues of their own: any rejection of the null hypothesis will be statistically valid, but they favor identifying differences similar to those seen before, and so may worsen gaps in performance between "well-represented" differences and rarer ones.

## Acknowledgments and Disclosure of Funding

FL and JL are supported by the Australian Research Council (ARC) under FL190100149. WX is supported by the Gatsby Charitable Foundation and EPSRC grant under EP/T018445/1. DJS is supported in part by the Natural Sciences and Engineering Research Council of Canada (NSERC) and the Canada CIFAR AI Chairs Program. FL would also like to thank Dr. Yanbin Liu and Dr. Yiliao Song for productive discussions.

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
