# A  Proofs and Additional Analysis

As a "warm-up" and because it is of independent interest, we will first study an adaptation algorithm which picks the single best kernel from the meta tasks:

**Definition 7** (Adaptation by choosing-one-best kernel). *With the set of base kernels $\{k_1, \ldots, k_N\}$, $\hat{k} = \arg\max_i \hat{J}_{\lambda^{ne}}(S_{\mathbb{P}}^{tr}, S_{\mathbb{Q}}^{tr}; k_i)$ is said to be the best kernel adaptation.*

Proposition 3 shows uniform convergence of $\hat{J}_\lambda$ for direct adaptation of a kernel class, whether a deep kernel or multiple kernel learning. For our analysis of choosing the best single kernel, however, we only need uniform convergence over a finite set, where we can obtain a slightly better rate.

**Lemma 8** (Generalization gap for choosing-one-best kernel adaptation). *Let $k_i$ be a set of base kernels, whose power criteria on the corresponding distributions are $J_i = J(\mathbb{P}, \mathbb{Q}; k_i)$, and let $s' = \min_{i \in [N]} \sigma_{\mathfrak{H}_1}^2(\mathbb{P}, \mathbb{Q}; k_i)$. Denote the regularized estimates of these values as $\hat{J}_i = \hat{J}_\lambda(S_{\mathbb{P}}, S_{\mathbb{Q}}; k_i)$, where $|S_{\mathbb{P}}| = |S_{\mathbb{Q}}| = m$ and $\lambda = m^{-1/3}$. Then, with probability at least $1 - \delta$,*

$$\max_{i \in [N]} |\hat{J}_i - J_i| \leq \zeta_m = \mathcal{O}\left( \frac{1}{s'^2 m^{1/3}} \left[ \frac{1}{s'} + \frac{1}{\sqrt{m}} + \sqrt{\log \frac{N}{\delta}} \right] \right). \tag{12}$$

*Proof.* To bound $\max_{i \in [N]} |\hat{J}_i - J_i|$, we consider high-probability bounds for concentration of $\hat{\eta}_\omega$ and $\hat{\sigma}_\omega^2$ with McDiarmid's inequality and a union bound, as developed within the proofs of Propositions 15 and 16 of Liu et al. [16]. With probability at least $1 - \delta$, we have

$$\max_{i \in [N]} |\hat{\eta}_i - \eta_i| \leq \frac{16\nu}{\sqrt{2m}} \sqrt{\log \frac{4N}{\delta}},$$

and

$$\max_{i \in [N]} |\hat{\sigma}_i^2 - \sigma_i^2| \leq 448 \sqrt{\frac{2}{m} \log \frac{4N}{\delta}} + \frac{1152\nu^2}{m}.$$

Then, taking $\sigma_{i,\lambda}^2 = \sigma_i^2 + \lambda$, we can decompose the worst-case generalization error as

$$
\begin{aligned}
\max_{i \in [N]} |\hat{J}_i - J_i| &= \max_{i \in [N]} \left| \frac{\hat{\eta}_i}{\hat{\sigma}_{i,\lambda}} - \frac{\eta_i}{\sigma_i} \right| \\
&\leq \max_{i \in [N]} \left| \frac{\hat{\eta}_i}{\hat{\sigma}_{i,\lambda}} - \frac{\hat{\eta}_i}{\sigma_{i,\lambda}} \right| + \max_{i \in [N]} \left| \frac{\hat{\eta}_i}{\sigma_{i,\lambda}} - \frac{\hat{\eta}_i}{\sigma_i} \right| + \max_{i \in [N]} \left| \frac{\hat{\eta}_i}{\sigma_i} - \frac{\eta_i}{\sigma_i} \right| \\
&\leq \max_{i \in [N]} \frac{|\hat{\eta}_i|}{\hat{\sigma}_{i,\lambda} \sigma_{i,\lambda} (\hat{\sigma}_{i,\lambda} + \sigma_{i,\lambda})} |\hat{\sigma}_i^2 - \sigma_i^2| + \max_{i \in [N]} \left| \frac{\hat{\eta}_i}{\sigma_{i,\lambda}} - \frac{\hat{\eta}_i}{\sigma_i} \right| + \max_{i \in [N]} \frac{|\hat{\eta}_i - \eta_i|}{\sigma_i} \\
&\leq \frac{4\nu}{s'^2 \sqrt{\lambda}} \max_{i \in [N]} |\hat{\sigma}_i^2 - \sigma_i^2| + \frac{2\nu}{s'^3} \lambda + \frac{1}{s'} \max_{i \in [N]} |\hat{\eta}_i - \eta_i|.
\end{aligned}
$$

Taking the upper bound on the kernel to be constant, in our case $\nu = 1$, the above equation reads

$$\max_{i \in [N]} |\hat{J}_i - J_i| = \mathcal{O}\left( \frac{1}{s'^2 \sqrt{\lambda}} \left[ \sqrt{\frac{1}{m} \log \frac{N}{\delta}} + \frac{1}{m} \right] + \frac{\lambda}{s'^3} + \frac{1}{s' \sqrt{m}} \sqrt{\log \frac{N}{\delta}} \right).$$

Taking the regularizer $\lambda = m^{-1/3}$ to achieve the best overall rate,

$$
\begin{aligned}
\max_{i \in [N]} |\hat{J}_i - J_i| &= \mathcal{O}\left( \frac{1}{s'^2 m^{1/3}} \left[ \sqrt{\log \frac{N}{\delta}} + \frac{1}{\sqrt{m}} \right] + \frac{1}{s'^3 m^{1/3}} + \frac{1}{s' \sqrt{m}} \sqrt{\log \frac{N}{\delta}} \right) \\
&= \mathcal{O}\left( \frac{1}{s'^2 m^{1/3}} \left[ \frac{1}{s'} + \sqrt{\log \frac{N}{\delta}} + \frac{1}{\sqrt{m}} \right] \right). \qquad \square
\end{aligned}
$$

Since the adaptation step is based on $m$ samples from the actual testing task, our generalization result is derived based on the sample size $m$. As explained in the main text, even though the sample size is

still small, the adaptation result benefits from a much better trained base kernel set, giving rise to large $s'$ compared to $s$ from directly training from the deep kernel parameters with $m$ samples.

Given this building block, we proceed to state and prove the choosing-one-best kernel adaptation, Theorem 10.

**Lemma 9.** *Let* $(\mathbb{P}, \mathbb{Q})$ *and* $(\mathbb{P}^i, \mathbb{Q}^i)$ *be two testing tasks which are* $\gamma$-*related (Definition 4), and let* $k^* \in \arg\max_{k \in \mathcal{K}} J(\mathbb{P}, \mathbb{Q}; k)$ *and* $k_i^* \in \arg\max_{k \in \mathcal{K}} J(\mathbb{P}^i, \mathbb{Q}^i; k)$. *Then*

$$|J(\mathbb{P}, \mathbb{Q}; k^*) - J(\mathbb{P}^i, \mathbb{Q}^i; k_i^*)| \le \gamma.$$

*Proof.* We know that $J(\mathbb{P}^i, \mathbb{Q}^i; k^*) \le J(\mathbb{P}^i, \mathbb{Q}^i; k_i^*)$ by the definition of $k_i^*$, and that $J(\mathbb{P}^i, \mathbb{Q}^i; k^*) \ge J(\mathbb{P}, \mathbb{Q}; k^*) - \gamma$ by $\gamma$-relatedness. Putting together we have

$$J(\mathbb{P}, \mathbb{Q}; k^*) - \gamma \le J(\mathbb{P}^i, \mathbb{Q}^i; k^*) \le J(\mathbb{P}^i, \mathbb{Q}^i; k_i^*),$$

and so $J(\mathbb{P}, \mathbb{Q}; k^*) - J(\mathbb{P}^i, \mathbb{Q}^i; k_i^*) \le \gamma$.

Similarly, we have

$$J(\mathbb{P}^i, \mathbb{Q}^i; k_i^*) - \gamma \le J(\mathbb{P}, \mathbb{Q}; k_i^*) \le J(\mathbb{P}, \mathbb{Q}; k^*)$$

and so $-\gamma \le J(\mathbb{P}, \mathbb{Q}; k^*) - J(\mathbb{P}^i, \mathbb{Q}^i; k_i^*)$. $\qquad\square$

**Theorem 10** (Adaptation by choosing one best base kernel). *Suppose we have* $N$ *meta-training tasks* $\{(\mathbb{P}^i, \mathbb{Q}^i)\}_{i \in [N]}$, *each with corresponding optimal kernels* $k_i^* \in \arg\max_{k \in \mathcal{K}} J(\mathbb{P}^i, \mathbb{Q}^i; k)$, *and learn kernels* $\hat{k}_i \in \arg\max_{k \in \mathcal{K}} \hat{J}_\lambda(S_{\mathbb{P}^i}, S_{\mathbb{Q}^i}; k)$ *based on* $n$ *samples in the setting of Proposition 3. Let* $(\mathbb{P}, \mathbb{Q})$ *be a test task from which we observe* $m$ *samples* $S_\mathbb{P}, S_\mathbb{Q}$. *Let* $j$ *be the index of a task* $(\mathbb{P}^j, \mathbb{Q}^j)$ *which is* $\gamma$-*related to* $(\mathbb{P}, \mathbb{Q})$. *Then, with probability at least* $1 - 2\delta$,

$$J(\mathbb{P}, \mathbb{Q}; k^*) - J(\mathbb{P}, \mathbb{Q}; \hat{k}) \le 2(\gamma + \xi_n^j + \zeta_m)$$

*where* $\xi_n^j$ *is the bound of Proposition 3 for* $(\mathbb{P}^j, \mathbb{Q}^j)$, *while* $\zeta_m$ *is the bound of Lemma 8 for* $(\mathbb{P}, \mathbb{Q})$.

*Proof.* We will assume that $(S_\mathbb{P}, S_\mathbb{Q})$ satisfies the uniform convergence condition of Lemma 8, and $(S_{\mathbb{P}^j}, S_{\mathbb{Q}^j})$ that of Proposition 3, which happens with probability at least $1 - 2\delta$. We use the decomposition

$$J(\mathbb{P}, \mathbb{Q}; k^*) - J(\mathbb{P}, \mathbb{Q}; \hat{k}) = \underbrace{J(\mathbb{P}, \mathbb{Q}; k^*) - J(\mathbb{P}^j, \mathbb{Q}^j; k_j^*)}_{(a)} + \underbrace{J(\mathbb{P}^j, \mathbb{Q}^j; k_j^*) - J(\mathbb{P}^j, \mathbb{Q}^j; \hat{k}_j)}_{(b)}$$

$$+ \underbrace{J(\mathbb{P}^j, \mathbb{Q}^j; \hat{k}_j) - J(\mathbb{P}, \mathbb{Q}; \hat{k}_j)}_{(c)} + \underbrace{J(\mathbb{P}, \mathbb{Q}; \hat{k}_j) - \hat{J}_\lambda(S_\mathbb{P}, S_\mathbb{Q}; \hat{k}_j)}_{(d)}$$

$$+ \underbrace{\hat{J}_\lambda(S_\mathbb{P}, S_\mathbb{Q}; \hat{k}_j) - \hat{J}_\lambda(S_\mathbb{P}, S_\mathbb{Q}; \hat{k})}_{(e)} + \underbrace{\hat{J}_\lambda(S_\mathbb{P}, S_\mathbb{Q}; \hat{k}) - J_\lambda(S_\mathbb{P}, S_\mathbb{Q}; \hat{k})}_{(f)}.$$

Lemma 9 upper-bounds (a) by $\gamma$, while Proposition 3 upper-bounds (b) by $2\xi_n^j$, and (c) is at most $\gamma$ by the definition of $\gamma$-relatedness. The terms (d) and (f) are each at most $\zeta_m$ by Lemma 8, while (e) is at most 0 by the definition of $\hat{k}$. The desired bound follows. $\qquad\square$

### Proof of Theorem 6 in the main text

*Proof.* Let $\beta^* \in \arg\max_{\beta \in \mathbb{R}_{\ge 0}^N} J(\mathbb{P}, \mathbb{Q}; \sum_i \beta_i^* \hat{k}_i)$, and then make the decomposition

$$J(\mathbb{P}, \mathbb{Q}; k^*) - J(\mathbb{P}, \mathbb{Q}; \hat{k}_{\hat{\beta}})$$

$$= \underbrace{J(\mathbb{P}, \mathbb{Q}; k^*) - J(\mathbb{P}, \mathbb{Q}; \hat{k}_j)}_{(i)} + \underbrace{J(\mathbb{P}, \mathbb{Q}; \hat{k}_j) - J(\mathbb{P}, \mathbb{Q}; \hat{k}_{\beta^*})}_{(ii)} + \underbrace{J(\mathbb{P}, \mathbb{Q}; \hat{k}_{\beta^*}) - J(\mathbb{P}, \mathbb{Q}; \hat{k}_{\hat{\beta}})}_{(iii)}.$$

Term (i) is identical to terms (a) through (c) of Theorem 10, and is upper-bounded by $2(\gamma + \xi_n^j)$ conditional only on the convergence event for $(S_{\mathbb{P}^j}, S_{\mathbb{Q}^j})$. Term (ii) is at most 0, since $\hat{k}_j$ corresponds to choosing the $j$th standard unit vector for $\beta$, so $\beta^*$ is at least as good as that choice of $\beta$. Finally, term (iii) is covered by Proposition 3, as in Proposition 8 of Liu et al. [16], giving an upper bound with probability $1 - \delta$ on $(S_\mathbb{P}, S_\mathbb{Q})$. $\qquad\square$

# B Experimental Details and Additional Experiments

## B.1 Datasets and Configurations

Figure 5 shows samples from *CIFAR-10* and *CIFAR-10.1*. *CIFAR-10.1* is available from https://github.com/modestyachts/CIFAR-10.1/tree/master/datasets (we use cifar10.1_v4_data.npy). This new test set contains $2,031$ images from TinyImages [54].

We implement all methods with Pytorch 1.1 (Python 3.8) using an NIVIDIA Quadro RTX 8000 GPU, and set up our experiments according to the protocol proposed by Liu et al. [16]. In the following, we demonstrate our configurations in detail. We run ME and SCF using the official code [5], and use Liu et al.'s implementations of most other tests. We use permutation test to compute $p$-values of C2ST-S, C2ST-L, MMD-O, MMD-D, AGT-KL, Meta-KL, Meta-MKL and all tests in Table 3. We set $\alpha = 0.05$ for all experiments. We use a deep neural network $g \circ \phi$ as the classifier in C2ST-S and C2ST-L, where $g$ is a two-layer fully-connected binary classifier, and $\phi$ is the feature extraction architecture also used in the deep kernels in MMD-D, AGT-KL, Meta-KL, Meta-MKL, and methods in Table 3 and Table 4.

For *HDGM*, $\phi$ is a five-layer fully-connected neural network. The number of neurons in hidden and output layers of $\phi$ are set to $3 \times d$, where $d$ is the dimension of samples. These neurons use softplus activations, $\log(1 + \exp(x))$. For *CIFAR*, $\phi$ is a convolutional neural network (CNN) with four convolutional layers and one fully-connected layer. The structure of the CNN follows the structure of the feature extractor in the discriminator of DCGAN [55] (see Figures 6 and 7 for the structure of $\phi$ in our tests, MMD-D, C2ST-S and C2ST-L). We randomly select data from two different classes to form the two samples ($n_i$ is 100) as meta-samples in CIFAR-10/CIFAR-100. Thus, there are $C_{10}^2$ and $C_{100}^2$ tasks when running Algorithm 1 on training sets of CIFAR-10 and CIFAR-100. For each task, we have 200 instances. Note that, for results on synthetic data, we repeat experiments 20 times to avoid the effects caused by the generation noise. DCGAN code is from https://github.com/eriklindernoren/PyTorch-GAN/blob/master/implementations/dcgan/dcgan.py.

We use the Adam optimizer [56] to optimize network and/or kernel parameters. Hyperparameter selection for ME, SCF, C2ST-S, C2ST-L, MMD-O and MMD-D follows Liu et al. [16]. In Algorithm 1, $\lambda$ is set to $10^{-8}$, and the update learning rate $\eta$ (line 2) is set to $0.8$, and the meta-update learning rate is set to $0.01$. Batch size is set to 10, and the maximum number of epoch is set to $1,000$. In line 6 in Algorithm 1, we use Adam optimizer with default hyperparameters. In line 1 in Algorithm 3, we adopt Adam optimizer with default hyperparameters and set learning rate to $0.01$. Besides, we use the algorithm from Algorithm 1 to initialize parameters in the optimization algorithm. To avoid the computational cost caused by the large number of meta-tasks, we randomly select 10 tasks in Meta-MKL rather than all $N$ tasks. Meanwhile, to ensure that we can get help from all tasks, we will use the algorithms outputted by Meta-KL to optimize the deep kernels (line 1 in Algorithm 3) in the selected 10 tasks. The algorithms outputted by Meta-KL are helpful to find the best deep kernel for each task. Note that we do not use dropout.

## B.2 Analysis of the Number of Tasks

We report the test power±standard error of Meta-KL and Meta-MKL when increasing the number of tasks $N$ from 20 to 150 in this subsection. Tables 5 and 6 show that the test power will increase in general when increasing $N$ from 20 to 150. When $m_{te} = 250$, the lowest test power appears when $N = 20$ (0.333 for Meta-KL and 0.459 for Meta-MKL), and the highest test power appears when $N = 150$ (0.771 for Meta-KL and 0.907 for Meta-MKL). This means that increasing the number of meta tasks will help improve the test power on the target task.

## B.3 Distinguishing CIFAR-10 or -100 from CIFAR-10.1 Using CIFAR-100-based Meta-tasks

In this subsection, we report results when meta-samples are generated by the training set of CIFAR-100 dataset, which are shown in Tables 7 and 8. It can be seen that our methods still have high test powers compared to previous methods. Besides, we can get higher test power on the task CIFAR-100 vs CIFAR-10.1 compared to results in Table 2, since meta-samples used here are closer to the target task. This phenomenon also appears in Section 5.3.

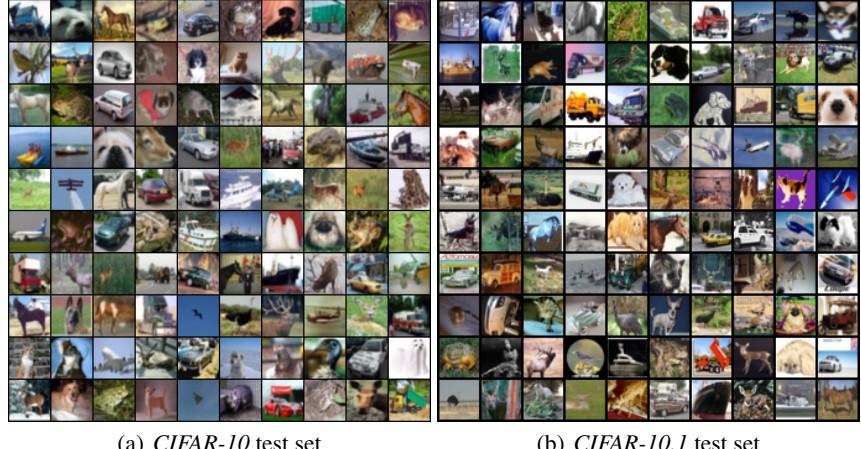

(a) *CIFAR-10* test set      (b) *CIFAR-10.1* test set

Figure 5: Images from *CIFAR-10* test set and the new *CIFAR-10.1* test set [53].

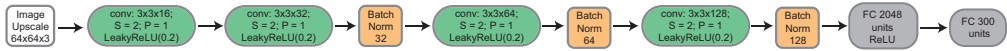

Figure 6: The structure of $\phi$ in our tests and MMD-D on *CIFAR*. The kernel size of each convolutional layer is 3; stride (S) is set to 2; padding (P) is set to 1. We do not use dropout in all layers. In the first layer, we will convert the *CIFAR* images from $32 \times 32 \times 3$ to $64 \times 64 \times 3$. Best viewed zoomed in.

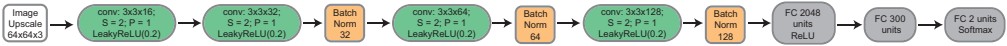

Figure 7: The structure of classifier $F$ in C2ST-S and C2ST-L on *CIFAR*. The kernel size of each convolutional layer is 3; stride (S) is set to 2; padding (P) is set to 1. We do not use dropout. Best viewed zoomed in.

Table 5: The test power of Meta-KL on the synthetic dataset given very limited training data ($\alpha = 0.05$, $m_{tr} = 50$) when increasing $N$ from 20 to 150. The $m_{te}$ represents number of samples when testing. Bold represents the highest mean per column.

| $m_{te}$ | 50 | 80 | 100 | 120 | 150 | 200 | 250 |
|---|---|---|---|---|---|---|---|
| $N = 20$ | 0.095±0.008 | 0.131±0.010 | 0.151±0.013 | 0.170±0.018 | 0.212±0.020 | 0.269±0.032 | 0.333±0.041 |
| $N = 50$ | 0.121±0.010 | 0.203±0.015 | 0.244±0.019 | 0.302±0.022 | 0.368±0.024 | 0.523±0.029 | 0.650±0.030 |
| $N = 80$ | 0.144±0.015 | 0.226±0.021 | 0.272±0.030 | 0.328±0.033 | 0.416±0.041 | 0.551±0.048 | 0.659±0.048 |
| $N = 100$ | 0.146±0.014 | 0.222±0.023 | 0.281±0.030 | 0.340±0.034 | 0.424±0.037 | 0.556±0.043 | 0.677±0.043 |
| $N = 120$ | 0.131±0.011 | 0.216±0.019 | 0.278±0.023 | 0.333±0.025 | 0.422±0.033 | 0.565±0.035 | 0.692±0.036 |
| $N = 150$ | **0.152±0.010** | **0.252±0.016** | **0.323±0.021** | **0.402±0.023** | **0.502±0.032** | **0.656±0.033** | **0.771±0.029** |

Table 6: The test power of Meta-MKL on the synthetic dataset given very limited training data ($\alpha = 0.05$, $m_{tr} = 50$) when increasing $N$ from 20 to 150. The $m_{te}$ represents number of samples when testing. Bold represents the highest mean per column.

| $m_{te}$ | 50 | 80 | 100 | 120 | 150 | 200 | 250 |
|---|---|---|---|---|---|---|---|
| $N = 20$ | 0.107±0.008 | 0.148±0.011 | 0.169±0.012 | 0.195±0.015 | 0.260±0.020 | 0.361±0.020 | 0.459±0.033 |
| $N = 50$ | 0.172±0.010 | 0.262±0.013 | 0.338±0.018 | 0.411±0.022 | 0.506±0.026 | 0.688±0.029 | 0.795±0.024 |
| $N = 80$ | 0.172±0.013 | 0.294±0.018 | 0.379±0.020 | 0.450±0.024 | 0.555±0.026 | 0.718±0.029 | 0.834±0.022 |
| $N = 100$ | 0.186±0.011 | 0.321±0.019 | 0.396±0.023 | 0.493±0.023 | 0.602±0.027 | 0.759±0.027 | 0.872±0.021 |
| $N = 120$ | 0.185±0.010 | 0.331±0.017 | 0.426±0.019 | 0.501±0.022 | 0.426±0.023 | 0.793±0.017 | 0.901±0.011 |
| $N = 150$ | **0.200±0.010** | **0.330±0.012** | **0.424±0.015** | **0.520±0.016** | **0.641±0.018** | **0.807±0.016** | **0.907±0.011** |

### B.4   Experiments regarding Closeness vs $\gamma$-relatedness

In this subsection, we introduce how to estimate the $\gamma$-relatedness between the target task $\mathcal{T} = (\mathbb{P}, \mathbb{Q})$ and the meta-tasks $\mathcal{T}^i = (\mathbb{P}^i, \mathbb{Q}^i)$.

Table 7: Test power of tests on CIFAR-10 vs CIFAR-10.1 given very limited training data ($\alpha = 0.05$, $m_{tr} = 100, 200$). The $m_{te}$ represents number of samples when testing. Bold represents the highest mean per column.

| Methods | $m_{tr} = 100$ | | | $m_{tr} = 200$ | | |
|---|---|---|---|---|---|---|
| | $m_{te} = 200$ | $m_{te} = 500$ | $m_{te} = 900$ | $m_{te} = 200$ | $m_{te} = 500$ | $m_{te} = 900$ |
| ME | 0.084±0.009 | 0.096±0.016 | 0.160±0.035 | 0.104±0.013 | 0.202±0.020 | 0.326±0.039 |
| SCF | 0.047±0.013 | 0.037±0.011 | 0.047±0.015 | 0.026±0.009 | 0.018±0.006 | 0.026±0.012 |
| C2ST-S | 0.059±0.009 | 0.062±0.007 | 0.059±0.007 | 0.052±0.011 | 0.054±0.011 | 0.057±0.008 |
| C2ST-L | 0.064±0.009 | 0.064±0.006 | 0.063±0.007 | 0.075±0.014 | 0.066±0.011 | 0.067±0.008 |
| MMD-O | 0.091±0.011 | 0.141±0.009 | 0.279±0.018 | 0.084±0.007 | 0.160±0.011 | 0.319±0.020 |
| MMD-D | 0.104±0.007 | 0.222±0.020 | 0.418±0.046 | 0.117±0.013 | 0.226±0.021 | 0.444±0.037 |
| AGT-KL | 0.172±0.035 | 0.465±0.044 | 0.812±0.033 | 0.143±0.021 | 0.438±0.073 | 0.836±0.065 |
| Meta-KL | 0.173±0.012 | 0.476±0.015 | 0.845±0.019 | 0.156±0.020 | 0.458±0.041 | 0.869±0.021 |
| Meta-MKL | **0.187±0.012** | **0.559±0.014** | **0.934±0.006** | **0.185±0.021** | **0.534±0.026** | **0.943±0.012** |

Table 8: Test power of tests on CIFAR-100 vs CIFAR-10.1 given very limited training data ($\alpha = 0.05$, $m_{tr} = 100, 200$). The $m_{te}$ represents number of samples when testing. Bold represents the highest mean per column.

| Methods | $m_{tr} = 100$ | | | $m_{tr} = 200$ | | |
|---|---|---|---|---|---|---|
| | $m_{te} = 200$ | $m_{te} = 500$ | $m_{te} = 900$ | $m_{te} = 200$ | $m_{te} = 500$ | $m_{te} = 900$ |
| ME | 0.211±0.020 | 0.459±0.045 | 0.751±0.054 | 0.236±0.033 | 0.512±0.076 | 0.744±0.090 |
| SCF | 0.076±0.027 | 0.132±0.050 | 0.240±0.095 | 0.136±0.036 | 0.245±0.066 | 0.416±0.114 |
| C2ST-S | 0.064±0.007 | 0.063±0.010 | 0.067±0.008 | 0.324±0.034 | 0.237±0.030 | 0.215±0.023 |
| C2ST-L | 0.089±0.010 | 0.077±0.010 | 0.075±0.010 | 0.378±0.042 | 0.273±0.032 | 0.262±0.023 |
| MMD-O | 0.214±0.012 | 0.624±0.013 | 0.970±0.005 | 0.199±0.016 | 0.614±0.017 | 0.965±0.006 |
| MMD-D | 0.244±0.011 | 0.644±0.030 | 0.970±0.010 | 0.223±0.016 | 0.627±0.031 | 0.975±0.006 |
| AGT-KL | 0.837±0.011 | **1.000±0.000** | **1.000±0.000** | 0.876±0.009 | **1.000±0.000** | **1.000±0.000** |
| Meta-KL | 0.938±0.016 | **1.000±0.000** | **1.000±0.000** | 0.962±0.005 | **1.000±0.000** | **1.000±0.000** |
| Meta-MKL | **0.966±0.006** | **1.000±0.000** | **1.000±0.000** | **0.985±0.005** | **1.000±0.000** | **1.000±0.000** |

**Estimation of $\gamma$-relatedness.** Let $S_{\mathbb{P}}$ and $S_{\mathbb{P}}$ be samples drawn from $\mathbb{P}$ and $\mathbb{Q}$, respectively, and let $S_{\mathbb{P}^i}$ and $S_{\mathbb{P}^i}$ be samples drawn from $\mathbb{P}^i$ and $\mathbb{Q}^i$, respectively. Then, we split $S_{\mathbb{P}}$ into $S_{\mathbb{P}}^{tr} \cup S_{\mathbb{P}}^{te}$, and $S_{\mathbb{Q}}$ into $S_{\mathbb{Q}}^{tr} \cup S_{\mathbb{Q}}^{te}$, and $S_{\mathbb{P}^i}$ into $S_{\mathbb{P}^i}^{tr} \cup S_{\mathbb{P}^i}^{te}$, and $S_{\mathbb{Q}^i}$ into $S_{\mathbb{Q}^i}^{tr} \cup S_{\mathbb{Q}^i}^{te}$. Let the deep kernel $k$ have the form (8). Next, following Definition 4 and [16], we find a kernel trying to achieve the maximum in $\gamma$ as

$$\hat{k} = \arg\max_k \left( \hat{J}(S_{\mathbb{P}}^{tr}, S_{\mathbb{Q}}^{tr}; k) - \hat{J}(S_{\mathbb{P}^i}^{tr}, S_{\mathbb{Q}^i}^{tr}; k) \right)^2. \tag{13}$$

Based on Definition 4, we can estimate the $\gamma_i$ between $\mathcal{T}$ and $\mathcal{T}^i$ as follows.

$$\hat{\gamma}_i = |\hat{J}(S_{\mathbb{P}}^{tr}, S_{\mathbb{Q}}^{tr}; \hat{k}) - \hat{J}(S_{\mathbb{P}^i}^{tr}, S_{\mathbb{Q}^i}^{tr}; \hat{k})|. \tag{14}$$

To try to avoid the local maximum during the the above maximizing process, we will repeat the above optimization procedure 10 times for estimating $\hat{\gamma}_i$. Namely, we have 10 values $\{\hat{\gamma}_{it}\}_{t=1}^{10}$ for $\hat{\gamma}_i$. Hence, the estimated $\gamma$ between $\mathcal{T}$ and $\{\mathcal{T}^i\}_{i=1}^N$ is set to $\hat{\gamma} = \min_i \max_t \hat{\gamma}_{it}$.

**Closeness vs $\gamma$-relatedness.** Given the target task $\mathcal{T}$ in synthetic datasets, in this experiment, we set $|S_{\mathbb{P}}^{tr}| = |S_{\mathbb{P}}^{te}| = |S_{\mathbb{Q}^i}^{tr}| = |S_{\mathbb{Q}^i}^{te}| = 4,000$ and define tasks $\boldsymbol{T}$ with closeness $C$ as

$$\boldsymbol{T}(C) = \{\mathcal{T}^i := (\mathbb{P}, \mathbb{Q}((0.6 - C) + 0.1 \times i/N))\}_{i=1}^N. \tag{15}$$

It is clear that $\boldsymbol{T}(0)$ will contain our target task $\mathcal{T}$ (i.e., the closeness is zero). Then, we estimate the $\gamma$-relatedness between the target task and $\boldsymbol{T}(C)$, where $C \in \{0.1, 0.2, 0.3, 0.4, 0.5\}$, and the results show that $\gamma \propto C$. Specifically, if we let $C$ be $0.1, 0.2, 0.3, 0.4, 0.5$, then the $\hat{\gamma}$ is $0.035, 0.067, 0.076, 0.104, 0.134$, respectively.