# OpenReview forum: "Meta Two-Sample Testing: Learning Kernels for Testing with Limited Data "
_NeurIPS.cc/2021/Conference — NeurIPS 2021 Poster_

### Official Review · Reviewer_Ye4V · 2021-06-30

**Rating:** 6
**Confidence:** 4

**Summary:**

State of the art kernel based two-sample tests operator as two-stage procedures. The available data is split, then first a kernel is optimized, and then an MMD test is performed on the other split of the data. With small available data, however, it is hard to reliably learn a good kernel in the first stage. This problem is particularly relevant if the class of kernels is very large, for example deep kernels.

Assuming abundant access to data from related tasks, the paper proposes to pretrain the kernel selection on the related tasks. This meta learning is argued to meaningfully constrain the space of available kernels. Thus for the actual task we can hope to reliably learn a good kernel even with little training data.

The proposed method “Meta multi-kernel learning” essentially learns a kernel for each related task. These then serve as candidate kernels. With the data from the actual problem they simply optimize a linear combination of the candidate kernels. The latter is a standard apprach. Usually, however, the candidate kernels are some default kernels, potentially chosen by expert knowledge. The propsed meta learning approach thus provides a principled and data-driven way to find suitable candidate kernels.

The authors provide some theoretical analysis and include synthetic and real-data experiments to support the benefits of Meta two-sample testing.


**Ethical Concerns:**

no concerns.

**Limitations And Societal Impact:**

As argued above the limitations of the approach should be stated more clearly.

**Main Review:**

I will first address the general dimensions and then include a list of comments and questions that I encourage the authors to address in order to improve the paper. Overall I can imagine the method a worthwhile contribution to the NeurIPS community. But not in the current state. I have some concerns regarding the objectiveness of the experiments. Since the paper is mainly supported based on the experiments, at the current stage I am reluctant to support acceptance.

**Originality**:
The proposed method is new. The theory is, however, largely build on the results of Liu et al [16]. Also it seems that the work Kirchler et al [14] deserves more explicit discussion. They also propose a method that potentially translates well to a related task. It would be good to discuss the similarities and differences to that method.

**Quality**:

* The theoretical claim (Theorem 7) largely follows from prior work of Liu et al [16] and seems correct.
* The provided code is not executable (missing requirements.txt or guidance on how to install required packages for example “freqopttest”). I was unable to reproduce the results or playing around with the hyperparameters.
* The empirical results only highlight cases where the proposed methods outperform existing methods. However, to more thoroughly judge the approach it would be good to understand (and include experiments) that show when exisiting methods are to be preferred. On the same note, what happens if the tasks are *not* related.
* I think the construction of the meta-tasks in the real application CIFAR should be thoroughly discussed in the main paper. Arguably this is the most critical step for a practicioner.


**Clarity**:

* The paper is generally clearly written and easy to understand for readers familiar with the challenges in two-sample testing.
* the presentation of Meta-KL is not so easy to understand (and the motivation and theoretical treatment not really provided). Do I understand correctly that in case of deep kernel, it essentially optimizes the parameter intialization? If so, maybe it is worth explicitly stating that.

**Significance**:
The paper defines a new task (improving two-sample tests with data from related tasks) and proposes a viable solution. However, in the current form I think it will be hard for practitioners to apply. It is not really clear how such situations look like where one has abundance of data of relevant related tasks. The construction of such related task should be discussed much more.


## Comments & Questions (please try to address in rebutal or resubmission):

1. CIFAR experiments:
    * For the meta learning samples from, say, P (the CIFAR1-10 dataset) are used. For a fair comparison, we should also allow the existing methods to see this data. This is indeed possible as at least some of them do not require balanced samples.
    * I assume that the differences between classes within CIFAR 10 is not the same as the difference between CIFAR-10 and CIFAR-10.1. Thus the Meta-Task are probably not perfect. Can it happen that we overfit to the meta-tasks? That is, if we use to much data and too aggressively train the meta kernels, does the performance decrease again?
2. Please include a requirement.txt in the code to make it easier to use. The implementation does not provide guidance on how to install required packages (eg. freqoptest). I was not able to run the code.
3. What is the motivation for including Meta-KL in the paper? For me Meta-MKL seems like a much more reasonable/natural approach and is much easier to grasp. The theory is only provided for the Meta-MKL, the experimental results show much better performance of Meta-MKL, and I think the implementation and application should be more straightforward. Based on this my impression is that the paper might be better without Meta-KL.
4. Defintion 4: is the maximization of \tilde{k} defined over all possible kernels? Or is the \gamma relatedness defined with respect to some class of kernels? Is it trivial that J(P||Q;k) is bounded for all k?
5. I am struggeling to exactly understand the messge of Theorem 7. What does the supremum mean here? Can you include an intuitive interpretation? Is this essentially guaranteeing, that with the proposed approach we will with high probability find a kernel such that J is close to the optimal J?
6. Line 196: I don’t think the claimed results on Meta-MKL is actually included in the appendix. Did I miss sth?
7. Figure 3: How is it possible that the results of Meta-MKL are significantly worse in panel c) than in panel b)? This raises some doubts about the implementation. I would expect that things can only be better if we have more data.
8. Table 2: Why does the power of MMD-D not improve when using $m_{tr}=200$ as opposed to $m_{tr}=100$. Again this raises some concerns about the implementation of the experiments.

## Minor comments (not relevant for score)

1. Algorithm 3: Why do you define the class of kernels with softmax weights. Simply saying we allow for a convex combination seems a bit easier (and is also used later on). don’t forget do adjust in line 159 accordingly.
2. Definition 5: The notation seems a bit confusing. What does the index $w$ indicate in the list of base kernels? Also in the maximizatin they are suddenly denotes as $k^{(i)}$. I don’t think $\lambda^{ne}$ was defined.
3. Definition 6: related to above. $w$ is used to both denote the linear weights in the combination as well as a subscript of the individual kernels.
4. line 219: Typo? I think $n$ and $N$ are the same?
5. Caption of Table 3: there is an “are” too much.


### ------ Update after Discussion phase --------
I updated the papers score from 5 to 6 - marginally above acceptance threshold.


**Time Spent Reviewing:**

8-10

---

> ### Author Response · Authors · 2021-08-06
> **Thanks for your comments – response**
>
> Thank you for your comments and effort in reviewing our work.
>
> > Also it seems that the work Kirchler et al [14] deserves more explicit discussion. They also propose a method that potentially translates well to a related task. It would be good to discuss the similarities and differences to that method.
>
> Kirchkler et al. [14] “pre-train” a kernel on a single related problem, but only suggest to use a linear “top-level” kernel – which Liu et al. [16] found performed notably worse than a deep kernel approach – and their theoretical analysis relies on strong assumptions about relatedness between their single related problem and their target (that each distribution is close in total variation to its test-time counterpart). Moreover, in practice they don’t optimize test power, but rather train a network on some hand-selected proxy task, e.g. a classification task, without providing any guarantees but rather relying on human intuition that the proxy features will be useful for the testing problem. We will add more explicit discussion of [14] to the manuscript in revision.
>
> > The provided code is not executable (missing requirements.txt or guidance on how to install required packages for example “freqopttest”). I was unable to reproduce the results or playing around with the hyperparameters.
>
> The package freqopttest is available from https://github.com/wittawatj/interpretable-test, but in fact we no longer use this package in our code and simply forgot to remove the import statements and lines 384-423 in utils.py. We will also add a requirements.txt file.
>
> > The empirical results only highlight cases where the proposed methods outperform existing methods. However, to more thoroughly judge the approach it would be good to understand (and include experiments) that show when exisiting methods are to be preferred. On the same note, what happens if the tasks are not related.
>
> Our experiment with increasing “closeness” (Figure 4) does demonstrate that test power decreases as the meta-tasks become less related (in this setup). As m increases, Figure 3 also “single-shot” testing methods beginning to catch up with the meta approaches. We can add (perhaps in the appendix) a case with high m and poor closeness, where existing single-shot methods may outperform or at least match our approach; this is a good idea.
>
> > I think the construction of the meta-tasks in the real application CIFAR should be thoroughly discussed in the main paper. Arguably this is the most critical step for a practicioner.
>
> Indeed, we will try to add detail and make this construction clearer in the main body.
>
> > the presentation of Meta-KL is not so easy to understand (and the motivation and theoretical treatment not really provided). Do I understand correctly that in case of deep kernel, it essentially optimizes the parameter intialization? If so, maybe it is worth explicitly stating that.
>
> Yes, Algorithm 1 as written only optimizes $\omega_\mathit{start}$, though it could easily also optimize e.g. the learning rate $\beta$. We will try to make this clearer and provide additional intuition for Meta-KL; it is basically an instance of the MAML framework. (We also, indeed, only provided theory for Meta-MKL, since it performs better in practice anyway.)
>
> > For the meta learning samples from, say, P (the CIFAR1-10 dataset) are used. For a fair comparison, we should also allow the existing methods to see this data. This is indeed possible as at least some of them do not require balanced samples.
>
> The advantage of this meta-testing setup in the “plain” CIFAR-10 vs CIFAR-10.1 problem is that none of the limited CIFAR-10.1 samples need to be “used up” for training purposes, although indeed the meta tasks are not exactly “correct” for our test task. Running e.g. MMD-D with all of CIFAR-10 vs all of CIFAR-10.1 is an interesting baseline that we will also run (in the next few days, and report back when done). This is, however, not possible with CIFAR-100, which provides a good example of related-but-not-identical data.
>
> > I assume that the differences between classes within CIFAR 10 is not the same as the difference between CIFAR-10 and CIFAR-10.1. Thus the Meta-Task are probably not perfect. Can it happen that we overfit to the meta-tasks? That is, if we use to much data and too aggressively train the meta kernels, does the performance decrease again?
>
> It does seem intuitively possible that we could overfit to the meta-tasks, particularly if the difference in the meta-tasks is too dissimilar to the difference in the target task. We didn’t observe this in our development, but will see if we can force this to happen in a contrived setup to get better intuition.
>
> > Please include a requirement.txt in the code to make it easier to use. The implementation does not provide guidance on how to install required packages (eg. freqoptest). I was not able to run the code.
>
> Unfortunately we don’t seem to be able to post updated code files at this point, but as mentioned above, you should be able to simply comment out the relevant import statements and lines 384-423 of utils.py. We will fix this and add a requirements.txt for the public version of the code.
>
> > What is the motivation for including Meta-KL in the paper?
>
> As mentioned to another reviewer, the “story” of our paper (and indeed what happened as we worked on it) was to first apply a standard MAML-type framework to the problem of meta-testing, see that it helped but not as much as we had hoped, and then come up with a more specialized algorithm for meta-testing that worked better. Without Meta-KL in the paper, we think that a reader familiar with the meta-learning literature would be thinking “why can’t you just do MAML here?” through the whole paper, so we wanted to answer that question. We will consider de-emphasizing Meta-KL in the main body of the paper, though, since indeed Meta-MKL is our primary proposed algorithm.
>
> > Defintion 4: is the maximization of \tilde{k} defined over all possible kernels? Or is the \gamma relatedness defined with respect to some class of kernels? Is it trivial that J(P||Q;k) is bounded for all k?
>
> We should indeed clarify that $\gamma$-relatedness implicitly depends on a kernel class, thanks. $J(\mathbb{P}, \mathbb{Q}; k)$ can be unbounded either if the MMD is infinite or the variance is zero. If either $k$ is bounded or the supports of $\mathbb{P}$ and $\mathbb{Q}$ are bounded (both are true in most of the examples we consider here), the MMD is necessarily finite. Zero variance happens only for either a trivial constant kernel (in which case the MMD is zero) or a kernel that perfectly distinguishes $\mathbb{P}$ and $\mathbb{Q}$, which we assume doesn’t happen in Theorem 7 and don’t think is particularly relevant in practice. (Also, $J_\lambda$ regularizes the denominator to be nonzero as well.)
>
> > I am struggeling to exactly understand the messge of Theorem 7.
>
> Theorem 7 is a uniform convergence type result; the supremum tells us that for any kernel considered during optimization, our estimate of the test power proxy $\hat{J}_\lambda$ is not too far from the true value $J$. Thus, indeed, the kernel that we choose by optimizing $\hat{J}_\lambda$ has $J$ value not too far from the optimal $J$. We will add a little discussion of this to the paper.
>
> > Line 196: I don’t think the claimed results on Meta-MKL is actually included in the appendix. Did I miss sth?
>
> Sorry, you’re totally right – this section ended up apparently being entirely removed from the appendix due to a last-minute editing error. Our theorem about Meta-MKL has the same form as Theorem 7, decomposing the worst-case generalization error of $\hat{J}$ into three parts. First, the $\beta_n$ term about the generalization gap of meta-training is the same as in Theorem 7. Second, the $\gamma$ term which relates the quality of kernels on the meta task to the target task: $\gamma’$ is essentially the same as Definition 4, but considering all convex combinations of the set of kernels. We know that $\gamma’$ is at most the $\gamma$ of Theorem 7, since the MKL can just choose to use a single kernel. Last, the $\tilde\beta$ term uses Proposition 8 of Liu et al. (2020) to bound the generalization gap of multiple kernel learning on $m$ samples, in the same way that our Theorem 7 uses our Lemma 8. We will of course add this back into the appendix (or the main body, if space allows). :)
>
> > Figure 3: How is it possible that the results of Meta-MKL are significantly worse in panel c) than in panel b)? This raises some doubts about the implementation. I would expect that things can only be better if we have more data.
> > Table 2: Why does the power of MMD-D not improve when using mtr=200 as opposed to mtr=100. Again this raises some concerns about the implementation of the experiments.
>
> Thanks for pointing out these oddities in the results. They seem to be just noise with the number of runs that we did for these figures – we reran the mtr={100, 150, 200} results with 100 replications instead of 10, and they now follow the order we’d expect. (Although the confidence bars didn’t quite overlap, there are many entries here to compare, so that seems to be due to a multiple comparisons problem.) It’s also worth noting that for MMD-D, we essentially just ran code directly from the authors [16]. We’ll revise to use 100 runs for all figures.
>
> > Minor comments
>
> Thanks for the typo fixes; line 219 and equation 11 were supposed to be a capital N. Definition 5 and related notation was indeed confusing; we’ve changed it in our draft to consistently be $k^{(i)}$ and $\hat{k}$, with no $\omega$ subscript. $\lambda^\mathit{ne}$ refers to the value of regularization used for $\hat{J}_\lambda$ in the adaptation step, as in Theorem 7, but that was unclear; we'll rephrase.

---

> > ### Comment · Reviewer_Ye4V · 2021-08-17
> > **Please help me understand the motivation for meta learning.**
> >
> > Dear authors,
> > after reading your response and the other reviews it is clearer to me, that my primary criticism is that I am not able to understand that meta tasks are actually available in practice.
> >
> > My concerns are completely independent of your technical contributions, i.e., I do not critizies the kernel optimization or anything like that. But since your paper is the first (that I know of) that proposes meta-learning for two-sample testing, I think it is absoulutely critical that this motivation is reasonable.
> >
> > So let's forget for a moment about kernels and most of what you said. Instead let us focus on one of the easiest examples of two-sample testing.
> > Assume that $P = \mathcal{N}(0_d,1_d)$ and $Q = \mathcal{N}(\mu,1_d)$ and you want to test whether $P=Q$.
> > Assume that $\mu= (1,0, \ldots, 0)^\top$. Then Meta tasks would only be reasonable if their difference is also mostly along the first dimension. But how to get such data???
> >
> > I can easily think of the situation where we additionally have unlabelled mixed samples from $P$ and $Q$, where still $\mu= (1,0, \ldots, 0)^\top$. In such a case, running PCA on the unlabelled data and then making a test only with the first principal component would greatly improve performance. This would not require any meta learning.
> >
> > But what if the covariance is still diagonal but has hudge variance in the first dimension (the one where the distributions actually differ). Then PCA would be misleading. Now having meta-tasks that still make me find that the first dimension is the one I should focus on would be great. But I really do not see how to come up with the appropriate labels for this tasks.
> >
> > Can you please try to give me intuition on how meta learning works here?

---

> > > ### Author Response · Authors · 2021-08-18
> > > **Re: Please help me understand the motivation for meta learning.**
> > >
> > > Thanks for your questions; we’ll try to help clarify things for you here (please follow up if we don’t succeed at that!), and will also try to put some more discussion of this motivation into the paper in revision.
> > >
> > > > Let us focus on one of the easiest examples of two-sample testing.
> > > > Assume that $P = \mathcal{N}(0_d,1_d)$ and $Q = \mathcal{N}(\mu,1_d)$ and you want to test whether $P=Q$.
> > > > Assume that $\mu= (1,0, \ldots, 0)^\top$. Then Meta tasks would only be reasonable if their difference is also mostly along the first dimension. But how to get such data???
> > >
> > > First, let’s note that this task is in fact quite difficult if $d$ is large and the number of samples is small. In that case, then, we’ll want some side information.
> > >
> > > As you suggested, if we have unlabelled data from a mixture of $P$ and $Q$, running a test only on the first principal component would solve this particular problem. It’s easy to come up with examples where such an approach entirely fails, however, if the overall variance of the data is in directions irrelevant to the difference between $P$ and $Q$ – e.g. $P = \\mathcal{N}\\left( \\begin{bmatrix}0 \\\\ 0\\end{bmatrix}, \\begin{bmatrix}1 & 0 \\\\ 0 & 100\\end{bmatrix}\\right)$ vs $Q = \\mathcal{N}\\left( \\begin{bmatrix}1 \\\\ 0\\end{bmatrix}, \\begin{bmatrix}1 & 0 \\\\ 0 & 100\\end{bmatrix}\\right)$.
> > >
> > > We thus suggest meta-testing as a framework that, when appropriate meta-tasks are available, can actually solve this problem. Let’s think about what good meta tasks for this problem might be.
> > >
> > > One extremely (in fact, unreasonably) helpful meta-task distribution would be, say,
> > > $$
> > > \\left\\{ \\left(
> > > \\mathcal{N}\\left( \\mathbf{0}\_d, \mathbf{I}\_d \\right),
> > > \\mathcal{N}\\left( \\begin{bmatrix} \lambda \\\\ 0 \\\\ \\vdots \\\\ 0 \\end{bmatrix}, \\mathbf{I}\_d \\right)
> > > \\right) \\right\\}
> > > $$
> > > where $\\lambda \\sim \\mathcal{Unif}(2, 3)$.
> > > Here we would quickly learn that only the first dimension matters, and get a good test. This would still work with an extremely skewed covariance matrix rather than $\\mathbf{I}_d$, as long as each meta-task has enough samples (unlike PCA-based approaches).
> > >
> > > But, it is of course unrealistic to assume that we will always have meta-tasks that so closely match the target task. Can our approach still work with more generic, less-directly related tasks?
> > >
> > > Yes. For instance, suppose that we have meta-tasks which (like the target task) are determined only by a difference in means. Then a linear kernel, so that the MMD becomes simply the difference in means between the distributions, would be a reasonable (though not optimal) answer, and it is reasonable to suspect that “aggregate” methods would learn a linear kernel. This is also a possible solution for meta-learning methods, and so we can hope that they would do at least as well.
> > >
> > > In fact, though, exploiting the adaptation available through meta-learning may let us get an even more powerful test. If our meta-tasks consist of various problems with a difference in means, then the learned kernel for each task (as in the meta-training phase of Meta-MKL) might be a linear projection onto the direction of that difference: $k(x, y) = x^T (\\mu\_P - \\mu\_Q) (\\mu_P - \\mu\_Q)^T y$. Then, at test time, if the difference in means is similar to a meta-task’s difference in means, Meta-MKL can just essentially select that kernel, looking at the difference in means. If not, it can use a linear combination of those kernels to best approximate the difference in means for the target task. (Meta-KL should also, hopefully, be able to find a similar solution, although it’s harder to reason about what it will do given its more generic form of adaptation.)
> > >
> > > This type of adaptation is exactly the motivation of meta-testing: we hope to find “what differences between distributions generally look like” on the meta-tasks, and then at test time, we can use our very limited data to search for differences in that more constrained set of options. Because the space of candidate rules is more limited, we can (hopefully) find a good rule with a much smaller number of data points.
> > >
> > > In practical situations, whether this setup is useful indeed depends on whether we can find an appropriate set of meta-tasks. There’s not too much we can say about that in very general terms; the issue is whether the “kind of” kernels that are good on the meta-tasks are also good on the target task. But we think this is often the case for two-sample tests: if we’re trying to detect changes in various populations, we can train on many related sub-populations, whether that’s slicing user data by demographics or epidemic data by geographic region.

---

> > > > ### Comment · Reviewer_Ye4V · 2021-08-19
> > > > **Re: Please help me understand the motivation for meta learning.**
> > > >
> > > > Thank you for the clarifications. I think by now I understand the setting quite well. It seems to boil down to the question of whether or not we are willed to assume that meta-tasks are at hand. Since I have limited experience with meta-learning I will consult the other reviewers on this question.
> > > >
> > > > But let me take the chance to illustrate what I meant by **overfitting** the meta tasks. In the example above let's assume that we have a meta task with a difference in means along the *second* dimension (while the true problem has the difference in means along the first).
> > > > Now you argued that learning the kernel could lead to a linear kernel (since that suffices). However, I think the optimal kernel for this meta task is $k_2^*(x,y) = x_2 \cdot y_2$ (and we should find this if we optimize the $J$ criterion with a lot of data, and accordingly decayed $\lambda$).
> > > >
> > > > Now clearly this kernel is completely useless for the true task, since the true task has a difference along the first dimension. Ergo we "overfitted" to the meta task.
> > > >
> > > > Does this change if we have many meta-tasks, say one for difference along each dimension? I don't think really. For each meta-task the optimal kernel will be to focus on a single dimension. So if the first one was not among the meta-tasks we are doomed. And it if was, than it's significance will not be so pronounced as likely we have also seen other distracting meta-tasks.
> > > >
> > > >
> > > >
> > > >
> > > > ------- Another comment regarding the $\gamma$ relatedness ------
> > > >
> > > > at the beginning of your comment you suggest the meta task
> > > > $$
> > > > \\left\\{ \\left(
> > > > \\mathcal{N}\\left( \\mathbf{0}\_d, \mathbf{I}\_d \\right),
> > > > \\mathcal{N}\\left( \\begin{bmatrix} \lambda \\\\ 0 \\\\ \\vdots \\ 0 \\end{bmatrix}, \\mathbf{I}\_d \\right)
> > > > \\right) \\right\\}
> > > > $$
> > > > Let us for simplicity use $\lambda =3$ instead of a uniform distribution.
> > > > Using definitions in Definition 4, for this meta task we have $\tilde{k}'(x,y) = x_1\cdot y_1$ (up to a constant) and $\tilde{k} = \tilde{k}'$, since the distributions differ in exactly the same way. This means the optimal kernels are exactly the same, but we have $3 J(P||Q; \tilde{k}) = J(P||Q; \tilde{k})$, and so $\gamma = 2$.
> > > >
> > > > I agree with what you wrote in line 169-170 (e.g., the meta task is simpler), but the lines before may require rethinking: The best kernel of both task are exactly the same - So from that argument I think the criterion should show that they are maximally related, but it shows $\gamma = 2$.
> > > >
> > > > Maybe there could be a more suitable criterion.

---

> > > > > ### Author Response · Authors · 2021-08-20
> > > > > **Response**
> > > > >
> > > > > For clarity, let's call the type of task at hand
> > > > > $$
> > > > > \\mathcal T_i: \\mathbb{P} = \\mathcal{N}(\\mathbf{0}\_d, \\mathbb{I}\_d) \text{ vs } \\mathbb{Q} = \\mathcal{N}(\mathbf{e}_i, \\mathbb{I}\_d)
> > > > > $$
> > > > > where $\mathbf{e}_i$ is the vector with 1 in the $i$th component and $0$s otherwise.
> > > > >
> > > > > If we imagine observing each of $\\{ \\mathcal T\_i \\}\_{i \in \\mathcal I}$ with $1 \notin \mathcal I$, with (essentially) infinitely many data points for each of these, then indeed (although we haven't quite proved this) we'd strongly expect that the learned kernels would be $\\{ \hat k\_i \\}\_{i \in \mathcal I}$ with $\hat k\_i(x, y) \propto x\_i y\_i$.
> > > > >
> > > > > In this particular situation, Meta-MKL would indeed be hopeless on the task $\\mathcal T_1$. This illustrates the "overfitting" idea suggested earlier.
> > > > > However:
> > > > > - We could alleviate the problem by always including a simple base kernel or two (e.g. linear and median-heuristic Gaussian kernel) in the set of candidate kernels.
> > > > > - If we didn't actually have infinitely many data points in each of the meta $\\mathcal T_i$, we'd expect most of the $\hat k_i$ to include some small $x_1 y_1$ component, and hence a linear combination should be capable of achieving a reasonable kernel.
> > > > > - Meta-KL, at least with a powerful enough network architecture, would not behave the same way. We can hope to learn an adaptation step which does roughly identify the one sparse component, which would then generalize to new $\\mathcal T_i$ perfectly. This may or may not be easy to identify with any particular kernel architecture, but it is exactly the kind of process aimed for in meta-testing (or general meta-learning).
> > > > >
> > > > > -----
> > > > >
> > > > > On $\gamma$-relatedness:
> > > > >
> > > > > First, just to establish notation more fully, call this task
> > > > > $$
> > > > > \\mathcal T^\\mu: \\mathbb{P} = \\mathcal{N}(\\mathbf{0}\_d, \\mathbb{I}\_d) \text{ vs } \\mathbb{Q} = \\mathcal{N}(\mu \mathbf{e}\_1, \\mathbb{I}\_d)
> > > > > ,$$
> > > > > and let $k(x, y) = x\_1 y\_1$. (Constants don't matter to $J$.)
> > > > > Then we have (see calculations below if interested, though it seems you basically figured this out)
> > > > > that
> > > > > $$J(\\mathcal T^\\mu; k) = \\frac{\\lvert \\mu \\rvert}{2\\sqrt{2}}.$$
> > > > >
> > > > > Plugging this into Definition 4, we see that the $\gamma$-relatedness between $\\mathcal T^\mu$ and $\\mathcal T^{\mu'}$ is indeed $\\big\\lvert \\lvert\mu\\rvert - \\lvert\\mu'\\rvert \\big\\rvert / (2 \\sqrt{2} )$, while ideally we would like them to be perfectly related.
> > > > >
> > > > > This makes our bound looser than it could be, and indeed gives an unsatisfying result in this case when the meta-tasks’ $\mu$s are all quite different from the target task’s. $\\gamma$-relatedness makes the most sense when the two problems are of “similar difficulty”; we will add some discussion of this to the final version of the paper.
> > > > >
> > > > > The reason we used this definition, rather than one based on $J(P \\| Q; k) - J(P \\| Q; k’)$, is that when we use empirical estimates $\\hat k$ and $\\hat{k’}$ rather than the population-optimal kernels $k$ and $k’$, we think additional unnatural assumptions about the “shape” of $J$ would be required to prove convergence bounds (that $J(P \\| Q; \\hat{k’})$ is close to $J(P \\| Q; k’)$). When the sample-learned $\hat{k’}$ performs well on $P$ and $Q$, however, the actual convergence gap will become small.
> > > > >
> > > > >
> > > > > **Calculations for $J$ on $\\mathcal T^\mu$, for completeness:**
> > > > > Here we have
> > > > > $\\operatorname{MMD}(\\mathcal T^\\mu; k) = \\lVert \\mathbb E X - \\mathbb E Y \\rVert = \\lvert \\mu \\rvert .$
> > > > > Recall that
> > > > > \\begin{align*}
> > > > > \\sigma_{\\mathcal H_1}^2(\\mathcal T^\\mu; k)
> > > > > &= 4 \\mathbb{E}[ H\_{ij} H\_{i \ell} ] - 4 \\mathbb{E}[ H\_{ij} ]^2
> > > > > \\\\
> > > > > H\_{ij}
> > > > > &= X\_{i,1} X\_{j,1} + Y\_{i,1} Y\_{j,1} - X\_{i,1} Y\_{j,1} - Y\_{i,1} X\_{j,1}
> > > > > \\\\&= (X\_{i,1} - Y\_{i,1}) (X\_{j,1} - Y\_{j,1})
> > > > > ,\\end{align*}
> > > > > and so one term of $\\sigma$ is
> > > > > $$
> > > > > \\mathbb{E}[H\_{ij}]
> > > > > = (0 - \mu) (0 - \mu) = \mu^2
> > > > > .$$
> > > > > For the other term,
> > > > > \\begin{align*}
> > > > > \\mathbb{E}[H\_{ij} H\_{i\ell}]
> > > > >     &= \\mathbb{E}\\left[
> > > > > (X\_{i,1} - Y\_{i,1}) (X\_{j,1} - Y\_{j,1})
> > > > > (X\_{i,1} - Y\_{i,1}) (X\_{\ell,1} - Y\_{\ell,1})
> > > > > \\right]
> > > > > \\\\&= \mu^2 \\mathbb{E}\\left[ (X\_{i,1} - Y\_{i,1})^2 \\right]
> > > > > \\\\&= \mu^2 (\mu^2 + 2)
> > > > > \\end{align*}
> > > > > so that
> > > > > $$
> > > > > \\sigma_{\\mathcal H_1}^2(\\mathcal T^\\mu; k)
> > > > > = 4 \mu^4 + 8 \mu^2 - 4 \mu^4
> > > > > = 8 \mu^2
> > > > > $$
> > > > > and
> > > > > $$J(\\mathcal T^\\mu; k) = \\frac{ \\mu^2 }{\\sqrt{8 \\mu^2}} = \\frac{\\lvert \\mu \\rvert}{2\\sqrt{2}}.$$

---

> > ### Comment · Reviewer_Ye4V · 2021-08-22
> > **Baseline - MMD-D with all of Cifar 10**
> >
> > > The advantage of this meta-testing setup in the “plain” CIFAR-10 vs CIFAR-10.1 problem is that none of the limited CIFAR-10.1 samples need to be “used up” for training purposes, although indeed the meta tasks are not exactly “correct” for our test task. **Running e.g. MMD-D with all of CIFAR-10 vs all of CIFAR-10.1 is an interesting baseline that we will also run** (in the next few days, and report back when done). This is, however, not possible with CIFAR-100, which provides a good example of related-but-not-identical data.
> >
> > Did you manage to run this experiment? (Instead of **all** Cifar-10, maybe it is better to run it such that MMD-D sees the same sample from Cifar-10 as did the meta-learning over all tasks together.)

---

> > > ### Author Response · Authors · 2021-08-22
> > > **Re: Baseline - MMD-D with all of Cifar 10**
> > >
> > > >Did you manage to run this experiment?
> > >
> > > Yes, we have managed this experiment (sorry for the waiting, our lab's server is quite busy recently) and booked a proper server. We are confident that all results will be gathered within 7 days!
> > >
> > > >Instead of all Cifar-10, maybe it is better to run it such that MMD-D sees the same sample from Cifar-10 as did the meta-learning over all tasks together.
> > >
> > > Sure, we can run this experiment and report the results when done.

---

> > > ### Author Response · Authors · 2021-08-27
> > > **Re: Baseline - MMD-D with all of Cifar 10**
> > >
> > > > Instead of all Cifar-10, maybe it is better to run it such that MMD-D sees the same sample from Cifar-10 as did the meta-learning over all tasks together.
> > >
> > > We have completed the corresponding experiments of MMD-D with all CIFAR10 (MMD-D w/ AC) that sees the same sample from CIFAR-10 as did the meta-learning over all tasks together. See the results (test power$\pm$standard error) below ($m_{tr}=100$). For your convenience, we also report the test power of MMD-D, AGT-KL, Meta-KL, and Meta-MKL below.
> > >
> > > | Methods |  $m_{te}=200$  | $m_{te}=500$   | $m_{te}=900$   |
> > > |-------|-----------------------|-------|-------|
> > > | MMD-D | 0.104$\pm$0.007  |  0.222$\pm$0.020 |  0.418$\pm$0.046  |
> > > | MMD-D w/ AC  | 0.134$\pm$0.010  |  0.325$\pm$0.028 |  0.745$\pm$0.049  |
> > > | AGT-KL     | 0.170$\pm$0.032  |  0.457$\pm$0.052 |  0.765$\pm$0.045  |
> > > | Meta-KL (ours)   | 0.245$\pm$0.010  |  0.671$\pm$0.026 |  0.959$\pm$0.013  |
> > > | Meta-MKL (ours)     | **0.277$\pm$0.016**  |  **0.728$\pm$0.020** |  **0.973$\pm$0.008**  |
> > >
> > > From the above results, we can see that the test power of MMD-D w/ AC clearly outperforms MMD-D since MMD-D w/ AC sees more CIFAR10 data in the training process (compared to MMD-D). It is also clear that our methods (Meta-KL and Meta-MKL) still perform much better than MMD-D w/ AC.  The MMD-D w/ AC is a very useful baseline, and we will include this baseline in our paper as well.
> > >
> > > Besides, we also run the MMD-D with all of CIFAR-10 vs all of CIFAR-10.1. Given the training-test ratio as 0.6, the test power of MMD-D is 0.836, which is a good reference of test power.
> > >
> > > Do you have more suggestions to improve the quality of our paper? It is glad to discuss our paper with you.

---

> > > > ### Comment · Reviewer_Ye4V · 2021-08-27
> > > > **Thanks for the additional experiments - will update my score**
> > > >
> > > > I thank the authors for reporting the additional experiments, which show that the improvement does not solely come from seeing more data from Cifar10. Instead the assigned labels for the meta-tasks indeed seem to help.
> > > >
> > > > I encourage the authors to dissect the origin of the advantages of the meta approaches in the revised version as good as possible.
> > > >
> > > > I will update my score to 6, but not above, as I still do not see how one will generally come up with labels for the meta-task.

---

> > > > > ### Author Response · Authors · 2021-08-27
> > > > > **Many thanks for your kind support!**
> > > > >
> > > > > Dear Reviewer Ye4V,
> > > > >
> > > > > Many thanks for your kind support!
> > > > >
> > > > > We will do as much as we can to dissect the origin of the advantages of the meta approaches in the revised version. We will also include our discussions here, which will improve the quality of our paper further.
> > > > >
> > > > > If you have more suggestions, please tell us. We want to improve our paper as much as possible.
> > > > >
> > > > > Best,
> > > > >
> > > > > Authors of Paper5204

---

### Official Review · Reviewer_D4sZ · 2021-07-14

**Rating:** 8
**Confidence:** 4

**Summary:**

It is a common issue to choose a proper kernel in learning problems or hypothesis testing problems. Learning a kernel is a good way to overcome this issue. However, the learning process needs a lot of data to ensure the effectiveness of the learned kernel. This paper considers another way to learn a kernel, i.e., learning a kernel from similar tasks rather than the target task, and implement this way in the kernel two-sample testing.

**Limitations And Societal Impact:**

Yes, the authors have adequately addressed the limitations and potential negative societal impact of their work.

**Main Review:**

Pros:

1.  The main contribution of this paper is to introduce meta-learning procedures to the field of kernel hypothesis testing. As known in the literature, kernel choice influences the learning performance/test power in the learning/hypothesis testing. To overcome this issue, researchers aim to split data into training and test sets and train a kernel using the training data. However, this solution has a natural drawback: it is not possible to use a lot of data to test, reducing the test power [r1]. The proposed idea overcomes this drawback wisely.

2. The proposed theory demonstrates what assumptions can ensure that a good kernel can be learned from similar tasks. This is a good start to apply meta-learning-like methods in the field of hypothesis testing.

3. Two-sample testing might become another task to test future meta-learning algorithms (e.g., few-shot learning, reinforcement learning). This might increase the significance of meta-learning algorithms, as two-sample testing is a fundamental problem in both statistic and machine learning.

Cons:

Q1. Although the proposed idea is novel, there are no new algorithms introduced in this paper. Algorithm 1 is mainly from MAML [r2]. I am not sure which part of MAML should be redesigned to fit the two-sample testing problem.

Q2. In the theory part, gamma-relatedness seems a very important definition. However, there are no illustrations to show what it means in synthetic data.

Q3. In Proposition 3, why is it reasonable to set lambda to be m^{1-/3}? How does the lambda influence the test power?

Q4. It is unclear why s’ should be greater than s. This is the key for Theorem 7.

Q5. Eq. (11) also measures similarity between two tasks. The difference between gamma-relatedness and Eq. (11) is unclear. In addition, hyperparameters should be analyzed.

[r1] Learning Kernel Tests Without Data Splitting, NeurIPS, 2020.

[r2] Model-agnostic meta-learning for fast adaptation of deep networks, ICML, 2017.


**Time Spent Reviewing:**

6

---

> ### Author Response · Authors · 2021-08-06
> **Thanks for your comments – response**
>
> Thank you for your comments and effort in reviewing our work.
>
> > Q1. Although the proposed idea is novel, there are no new algorithms introduced in this paper. Algorithm 1 is mainly from MAML [r2]. I am not sure which part of MAML should be redesigned to fit the two-sample testing problem.
>
> Algorithm 1 indeed closely follows a MAML-type learning scheme; Algorithm 3, though, is different, and to our knowledge novel in this setting (though of course building on existing work). In some sense the “story” of our paper is an attempt to straightforwardly apply MAML to this problem, but then realizing that a somewhat different approach performs better.
>
> > Q2. In the theory part, gamma-relatedness seems a very important definition. However, there are no illustrations to show what it means in synthetic data.
>
> Thanks for this suggestion; we will do some empirical estimation of how $\gamma$ changes with respect to $N$ and relates to (11) in the synthetic example, and add this to the revised draft.
>
> > Q3. In Proposition 3, why is it reasonable to set lambda to be m^{1-/3}? How does the lambda influence the test power?
>
> The choice $\lambda = \Theta\left( m^{-1/3} \right)$ optimizes the resulting bound; it is the same choice made by [16, Theorem 6], and arises from the form of the objective $\hat{J}_\lambda$. Choosing too high of a $\lambda$ would mean that the optimization essentially ignores the variance term in the denominator of $J$ (causing test power to suffer), while too low of a $\lambda$ would mean that the optimization might choose kernels that aren't actually very powerful but happened to have a really small estimate for the variance.
>
> > Q4. It is unclear why s’ should be greater than s. This is the key for Theorem 7.
> $\newcommand{\baromega}{\bar{\Omega}_s}$
>
> In the setup of Theorem 7, each of the candidate kernels $k_\omega^i$ is chosen from $\baromega$,
> which guarantees that for each $i$, $\sigma_{\mathcal{H}_1}(\mathbb{P}, \mathbb{Q}; k_\omega^i) \ge s$.
> Thus $s’$, which is the minimum of those terms, is also at least $s$.
>
> In practice, we expect that $s’ \gg s$, because $s$ is typically chosen to be some quite-small lower bound over a continuous space of kernels, while $s’$ is only the minimum of $N$ particular points in that space.
>
> > Q5. Eq. (11) also measures similarity between two tasks. The difference between gamma-relatedness and Eq. (11) is unclear. In addition, hyperparameters should be analyzed.
>
> Indeed, (11) and $\gamma$-relatedness both measure similarity between tasks, but we used (11) in those experiments because it is more intuitive in this particular problem. As mentioned above, we will add a plot relating an empirical estimate of $\gamma$ with closeness.
>
> In terms of hyperparameters, we mostly just followed the decisions of Liu et al. [16] for network parameters. We will also add an ablation study considering altering, e.g., $n_\mathit{steps}$ inside Algorithm 1.

---

> > ### Comment · Reviewer_D4sZ · 2021-08-17
> > **Thanks for the response**
> >
> > Thanks for addressing my concerns. Overall, I think this paper is novel. However, it would be better to implement the "decreasing N" experiment as another reviewer suggested.

---

> > > ### Author Response · Authors · 2021-08-19
> > > **Experiments regarding "decreasing N"**
> > >
> > > Thanks for the suggestion.
> > >
> > > We have completed the experiments regarding "decreasing N". Except for decreasing N, we also increase N to see how the test power changes (sorry for the waiting, our lab's server is quite busy these days). See the results (test power$\pm$standard error) of Meta-KL below ($m_{tr}=50$). Since we repeat each experiment $20$ times (according to comments from Reviewer Ye4V, we need to repeat more times to get more reasonable power. In the original version, we repeat each experiment $10$ times), the results reported here might be slightly different from results reported in Figure 3(a). We will report the test power of Meta-MKL when done.
> > >
> > > |  $m_{te}$  | 50   | 80   | 100  | 120    | 150    | 200 | 250|
> > > |-----------------------|-------|-------|-------|-------|-------|-------|-------|
> > > | $N=20$     | 0.095$\pm$0.008  |  0.131$\pm$0.010 |  0.151$\pm$0.013  |  0.170$\pm$0.018 |  0.212$\pm$0.020  |  0.269$\pm$0.032  |  0.333$\pm$0.041 |
> > > | $N=50$     | 0.121$\pm$0.010  |  0.203$\pm$0.015 |  0.244$\pm$0.019  |  0.302$\pm$0.022 |  0.368$\pm$0.024  |  0.523$\pm$0.029  |  0.650$\pm$0.030 |
> > > | $N=80$     | 0.144$\pm$0.015  |  0.226$\pm$0.021 |  0.272$\pm$0.030  |  0.328$\pm$0.033 |  0.416$\pm$0.041  |  0.551$\pm$0.048  |  0.659$\pm$0.048 |
> > > | $N=100$   | 0.146$\pm$0.014  |  0.222$\pm$0.023 |  0.281$\pm$0.030  |  0.340$\pm$0.034 |  0.424$\pm$0.037  |  0.556$\pm$0.043  |  0.677$\pm$0.043 |
> > > | $N=120$   | 0.131$\pm$0.011  |  0.216$\pm$0.019 |  0.278$\pm$0.023  |  0.333$\pm$0.025 |  0.422$\pm$0.033  |  0.565$\pm$0.035  |  0.692$\pm$0.036 |
> > > | $N=150$   | 0.152$\pm$0.010  |  0.252$\pm$0.016 |  0.323$\pm$0.021  |  0.402$\pm$0.023 |  0.502$\pm$0.032  |  0.656$\pm$0.033  |  0.771$\pm$0.029 |
> > >
> > > From the above results, we can see that the test power will increase in general when increasing $N$ from $20$ to $150$. The lowest test power appears when $N=20$ (0.333), and the highest test power appears when $N=150$ (0.771). This means that increasing the number of meta tasks will help improve the test power on the target task.

---

> > > > ### Comment · Reviewer_D4sZ · 2021-08-27
> > > > **Thanks for providing the additional experiments**
> > > >
> > > > I would like to thank the authors' efforts to improve the quality of this paper further. These experiments will alleviate many concerns. I am satisfied with the paper after those changes and would like to support this paper.

---

> > > ### Author Response · Authors · 2021-08-27
> > > **Follow-up Experimental Results**
> > >
> > > Thanks for the suggestion.
> > >
> > > We have completed the corresponding experiments of Meta-MKL. See the results (test power$\pm$standard error) below ($m_{tr}=50$). Since we repeat each experiment $20$ times (according to comments from Reviewer Ye4V, we need to repeat more times to get more reasonable power. In the original version, we repeat each experiment $10$ times), the results reported here might be slightly different from results reported in Figure 3(a).
> > >
> > > |  $m_{te}$  | 50   | 80   | 100  | 120    | 150    | 200 | 250|
> > > |-----------------------|-------|-------|-------|-------|-------|-------|-------|
> > > | $N=20$     | 0.107$\pm$0.008  |  0.148$\pm$0.011 |  0.169$\pm$0.012  |  0.195$\pm$0.015 |  0.260$\pm$0.020  |  0.361$\pm$0.029  |  0.459$\pm$0.033 |
> > > | $N=50$     | 0.172$\pm$0.010  |  0.262$\pm$0.013 |  0.338$\pm$0.018  |  0.411$\pm$0.022 |  0.506$\pm$0.026  |  0.688$\pm$0.029  |  0.795$\pm$0.024 |
> > > | $N=80$     | 0.172$\pm$0.013  |  0.294$\pm$0.018 |  0.379$\pm$0.020  |  0.450$\pm$0.024 |  0.555$\pm$0.026  |  0.718$\pm$0.029  |  0.834$\pm$0.022 |
> > > | $N=100$   | 0.186$\pm$0.011  |  0.321$\pm$0.019 |  0.396$\pm$0.023  |  0.493$\pm$0.023 |  0.602$\pm$0.027  |  0.759$\pm$0.027  |  0.872$\pm$0.021 |
> > > | $N=120$   | 0.185$\pm$0.010  |  0.331$\pm$0.017 |  0.426$\pm$0.019  |  0.501$\pm$0.022 |  0.626$\pm$0.023  |  0.793$\pm$0.017  |  0.901$\pm$0.011 |
> > > | $N=150$   | 0.200$\pm$0.010  |  0.330$\pm$0.012 |  0.424$\pm$0.015  |  0.520$\pm$0.016 |  0.641$\pm$0.018  |  0.807$\pm$0.016  |  0.907$\pm$0.011 |
> > >
> > > From the above results, we can see that the test power will increase in general when increasing $N$ from $20$ to $150$. The lowest test power appears when $N=20$ (0.459), and the highest test power appears when $N=150$ (0.907). This means that increasing the number of meta tasks will help improve the test power on the target task.

---

### Official Review · Reviewer_v3Mq · 2021-07-14

**Rating:** 6
**Confidence:** 4

**Summary:**

The paper introduces a new way of doing 2 sample testing in the setting where we are given multiple related 2 sample testing tasks and in the setting where we only have limited data. They propose 2 novel methods that allow them to use related information from the "meta training" tasks in order to learn the "best" kernel for 2 sample testing for a new unseen task. The main idea they are trying to stress is how to do 2 sample testing, in the setting where one is only given little data. They show that their 2 proposed method beats standard 2 sample tests by leveraging related tasks. They also provide a theoretical justification and bounds on their method.

**Limitations And Societal Impact:**

No societal impact

**Main Review:**

Reasons to accept:
-  Propose a new scenario for 2 sample testing in the meta learning setting,  which to the best of my knowledge has not been done before.
-  Propose a theoretical bound on the error for the meta learning setup (I am not familiar with this and hence i will rely on the other reviewers comments on this, however it does look promising)
-  Apply a simple MAML style algorithm to learn from related task and show significant improvements on synthetic data
-  Use an ensemble style algorithm to further improve their proposed method.
-  Analyse their algorithm on a synthetic dataset where closeness of the meta learning tasks can be measured

Reason to reject and clarification:
-  How does your algorithm depend on N? It would be interested to see if a decrease in N actually decreases the performance of the algorithm and if so by how much. N=100 seems rather arbitrary on the synthetic datasets and hence I would like the authors to comment on this.
-  What is the N in the CIFAR example? I tried to look up the experiment in the code provided but did not find the relevant info, neither in the appendix nor in the code base (I apologise if i missed this detail). The way i understand it m_tr means the number of samples in the meta training task and not the amount of meta training tasks. In line 219 you mention N for the synthetic datasets but not for the CIFAR dataset
-  You also mention that one could use the idea of having little data in the scenario of medical imaging in your introduction line 28-33. However your "real" example is only on CIFAR dataset and I do not see a straight forward way to apply it on some more real scenarios. If it is straight forward i would ask the authors to provide proof that it actually works on other that CIFAR datasets and maybe consider the medical imaging one that they described in the introduction.
-  This is might main criticism of this paper as it contains interesting ideas but they do not seem to have any application to test that the method works outside standardised synthetic datasets such as CIFAR. (They do create a new task that has not been done before) If what you said in the intro is true, to motivate your work, why not use that example in the experiments? You might not be able to have access to that data but in that case you would only have 3-5 images per patient no? and not 100+ as in your experiments. In these cases how does you algorithm work? What i want to say is, could you please provide extra motivation why CIFAR example is enough to convince me that it would work on "real" medical data as well?

-  I will also be checking out the code and in particular i am curious why no dropout was used in the NN and how the learning rates as well as the number of epochs affect the performance. I assume you used a validation set to determine the num_epochs etc etc. Please comment on the above.

Overall i think the paper is well written and presents a new avenue for meta learning. You provided theoretical bounds as well as good synthetic experiments in terms of testing power. I am more than happy to raise my score if the above concerns have been addressed. And will be looking forward to your rebuttal.

**Time Spent Reviewing:**

4

---

> ### Author Response · Authors · 2021-08-06
> **Thanks for your comments – response**
>
> Thank you for your comments and effort in reviewing our work.
>
> > How does your algorithm depend on N? It would be interested to see if a decrease in N actually decreases the performance of the algorithm and if so by how much. N=100 seems rather arbitrary on the synthetic datasets and hence Iwould like the authors to comment on this.
>
> We indeed didn’t run experiments examining in detail the performance as a function of the number of meta-tasks $N$, though of course if $N$ gets very small then performance will suffer – in particular, this will imply that the $\gamma$-relatedness quantity of our theoretical analysis will likely become large (as we have fewer chances for meta tasks to be similar), making our upper bound worse. We will add an experiment along these lines to the appendix in revision.
>
> > What is the N in the CIFAR example?
>
> We did mention N in the CIFAR example (though it’s a little hidden) around line 465, Appendix B.2 (second paragraph): we have N = 10 choose 2 = 45 for the CIFAR-10 training task, and N = 100 choose 2 = 4,950 for the CIFAR-100 training task. We’ll mention this more prominently.
>
> > You also mention that one could use the idea of having little data in the scenario of medical imaging in your introduction line 28-33. However your "real" example is only on CIFAR dataset and I do not see a straight forward way to apply it on some more real scenarios. If it is straight forward i would ask the authors to provide proof that it actually works on other that CIFAR datasets and maybe consider the medical imaging one that they described in the introduction.
> >
> > This is might main criticism of this paper as it contains interesting ideas but they do not seem to have any application to test that the method works outside standardised synthetic datasets such as CIFAR. (They do create a new task that has not been done before) If what you said in the intro is true, to motivate your work, why not use that example in the experiments? You might not be able to have access to that data but in that case you would only have 3-5 images per patient no? and not 100+ as in your experiments. In these cases how does you algorithm work? What i want to say is, could you please provide extra motivation why CIFAR example is enough to convince me that it would work on "real" medical data as well?
>
> It’s worth noting that distinguishing CIFAR 10 from 10.1 is not such an easy problem – Recht et al. [38] essentially tried to do so with a classifier two-sample test but were unsuccessful. Applications to problems such as the hypothetical medical imaging example would follow the same lines as shown here, but introduce many complications of their own in data processing that are quite independent from the contribution of this paper, and so we leave doing so to future work more focused on the medical application.
>
> It’s also worth noting that the number of images per patient is not the relevant number – rather, we’d be concerned with the number of images (say) from all patients in a given hospital with COVID-19 in a given month, which would presumably be substantially more than 3.
>
> > I will also be checking out the code and in particular i am curious why no dropout was used in the NN and how the learning rates as well as the number of epochs affect the performance. I assume you used a validation set to determine the num_epochs etc. Please comment on the above.
>
> Our architectural decisions and learning rates, number of epochs, etc followed Liu et al. [16], where they did not use the dropout; we didn’t do extensive parameter tuning here.
>
> > Overall i think the paper is well written and presents a new avenue for meta learning. You provided theoretical bounds as well as good synthetic experiments in terms of testing power. I am more than happy to raise my score if the above concerns have been addressed. And will be looking forward to your rebuttal.
>
> Thanks for your valuable comments and support! We would be glad to discuss further if we haven’t adequately addressed any of your points, or any other thoughts you have.

---

> > ### Comment · Reviewer_v3Mq · 2021-08-16
> > **Reply**
> >
> > > We indeed didn’t run experiments examining in detail the performance as a function of the number of meta-tasks $N$, though of course if $N$ gets very small then performance will suffer – in particular, this will imply that the $\gamma$-relatedness quantity of our theoretical analysis will likely become large (as we have fewer chances for meta tasks to be similar), making our upper bound worse. We will add an experiment along these lines to the appendix in revision.
> >
> > I would urge the authors to implement these and report here as this experiment would clear all the potential of meta overfitting as mentioned in other reviews. Could the authors also please upload the command line link they used to generate these.
> >
> > > We did mention N in the CIFAR example (though it’s a little hidden) around line 465, Appendix B.2 (second paragraph): we have N = 10 choose 2 = 45 for the CIFAR-10 training task, and N = 100 choose 2 = 4,950 for the CIFAR-100 training task. We’ll mention this more prominently.
> >
> > Thanks that is my bad.
> >
> > > It’s worth noting that distinguishing CIFAR 10 from 10.1 is not such an easy problem – Recht et al. [38] essentially tried to do so with a classifier two-sample test but were unsuccessful. Applications to problems such as the hypothetical medical imaging example would follow the same lines as shown here, but introduce many complications of their own in data processing that are quite independent from the contribution of this paper, and so we leave doing so to future work more focused on the medical application.
> >
> > Could the authors link me to the exact paragraph they are referring to in [38] for they claim? I am still a little skeptical about the general experimental setup, but I will consult with the other reviewers on this matter. If they believe this is a fair setup then I have nothing to object.
> >
> >
> > I will also be waiting until reviewer Ye4V replies before I update my score.

---

> > > ### Author Response · Authors · 2021-08-16
> > > **Reference inside Recht et al.**
> > >
> > > We’re referring to Appendix B.2.8: https://arxiv.org/pdf/1902.10811.pdf#page30 (we’ll add an explicit pointer to the section when we reference it; it is a very long paper). They don’t use the language of two-sample testing, but their statement
> > >
> > > > Although the models performed slightly better than random chance, the confidence intervals (95% Clopper Pearson) still overlap with 50% accuracy.
> > >
> > > implies that a sign-based C2ST with $\alpha = 0.05$ would not reject the null hypothesis.
> > >
> > > (We don’t have the changing-$N$ experiment results yet, but will try to run those soon.)

---

> > > ### Author Response · Authors · 2021-08-19
> > > **Experiments regarding "decreasing N"**
> > >
> > > >I would urge the authors to implement these and report here as this experiment would clear all the potential of meta overfitting as mentioned in other reviews.
> > >
> > > Thanks for the suggestion.
> > >
> > > We have completed the experiments regarding "decreasing N". Except for decreasing N, we also increase N to see how the test power changes (sorry for the waiting, our lab's server is quite busy these days). See the results (test power$\pm$standard error) of Meta-KL below ($m_{tr}=50$). Since we repeat each experiment $20$ times (according to comments from Reviewer Ye4V, we need to repeat more times to get more reasonable power. In the original version, we repeat each experiment $10$ times), the results reported here might be slightly different from results reported in Figure 3(a). We will report the test power of Meta-MKL when done.
> > >
> > > |  $m_{te}$  | 50   | 80   | 100  | 120    | 150    | 200 | 250|
> > > |-----------------------|-------|-------|-------|-------|-------|-------|-------|
> > > | $N=20$     | 0.095$\pm$0.008  |  0.131$\pm$0.010 |  0.151$\pm$0.013  |  0.170$\pm$0.018 |  0.212$\pm$0.020  |  0.269$\pm$0.032  |  0.333$\pm$0.041 |
> > > | $N=50$     | 0.121$\pm$0.010  |  0.203$\pm$0.015 |  0.244$\pm$0.019  |  0.302$\pm$0.022 |  0.368$\pm$0.024  |  0.523$\pm$0.029  |  0.650$\pm$0.030 |
> > > | $N=80$     | 0.144$\pm$0.015  |  0.226$\pm$0.021 |  0.272$\pm$0.030  |  0.328$\pm$0.033 |  0.416$\pm$0.041  |  0.551$\pm$0.048  |  0.659$\pm$0.048 |
> > > | $N=100$   | 0.146$\pm$0.014  |  0.222$\pm$0.023 |  0.281$\pm$0.030  |  0.340$\pm$0.034 |  0.424$\pm$0.037  |  0.556$\pm$0.043  |  0.677$\pm$0.043 |
> > > | $N=120$   | 0.131$\pm$0.011  |  0.216$\pm$0.019 |  0.278$\pm$0.023  |  0.333$\pm$0.025 |  0.422$\pm$0.033  |  0.565$\pm$0.035  |  0.692$\pm$0.036 |
> > > | $N=150$   | 0.152$\pm$0.010  |  0.252$\pm$0.016 |  0.323$\pm$0.021  |  0.402$\pm$0.023 |  0.502$\pm$0.032  |  0.656$\pm$0.033  |  0.771$\pm$0.029 |
> > >
> > > From the above results, we can see that the test power will increase in general when increasing $N$ from $20$ to $150$. The lowest test power appears when $N=20$ (0.333), and the highest test power appears when $N=150$ (0.771). This means that increasing the number of meta tasks will help improve the test power on the target task.
> > >
> > > >Could the authors also please upload the command line link they used to generate these.
> > >
> > > The data generation process can be seen in lines 90 - 103 in main_metaKL.py in the submitted code. In the above experiments, we change the variable “num_meta_tasks” from $20$ to $150$.

---

> > > > ### Comment · Reviewer_v3Mq · 2021-08-23
> > > > **Thanks for the clarifications**
> > > >
> > > > I thank the authors for their additional experiments which further strengthen this submission.
> > > >
> > > > Even though i still believe that the experiments are limited in a sense that it is only on synthetic datasets, I see the merits as being the first meta learning paper for 2 sample testing.
> > > >
> > > > Hence I will increase my score to 6.
> > > >
> > > > Best

---

> > > > > ### Author Response · Authors · 2021-08-27
> > > > > **Many thanks for your kind support!**
> > > > >
> > > > > Dear Reviewer v3Mq,
> > > > >
> > > > > Many thanks for your kind support!
> > > > >
> > > > > Do you have more suggestions to improve the quality of our paper? We are glad to discuss our paper with you.
> > > > >
> > > > > Best,
> > > > >
> > > > > Authors of Paper5204

---

> > > ### Author Response · Authors · 2021-08-27
> > > **Follow-up Experimental Results**
> > >
> > > >I would urge the authors to implement these and report here as this experiment would clear all the potential of meta overfitting as mentioned in other reviews.
> > >
> > > Thanks for the suggestion.
> > >
> > > We have completed the corresponding experiments of Meta-MKL. See the results (test power$\pm$standard error) below ($m_{tr}=50$). Since we repeat each experiment $20$ times (according to comments from Reviewer Ye4V, we need to repeat more times to get more reasonable power. In the original version, we repeat each experiment $10$ times), the results reported here might be slightly different from results reported in Figure 3(a).
> > >
> > > |  $m_{te}$  | 50   | 80   | 100  | 120    | 150    | 200 | 250|
> > > |-----------------------|-------|-------|-------|-------|-------|-------|-------|
> > > | $N=20$     | 0.107$\pm$0.008  |  0.148$\pm$0.011 |  0.169$\pm$0.012  |  0.195$\pm$0.015 |  0.260$\pm$0.020  |  0.361$\pm$0.029  |  0.459$\pm$0.033 |
> > > | $N=50$     | 0.172$\pm$0.010  |  0.262$\pm$0.013 |  0.338$\pm$0.018  |  0.411$\pm$0.022 |  0.506$\pm$0.026  |  0.688$\pm$0.029  |  0.795$\pm$0.024 |
> > > | $N=80$     | 0.172$\pm$0.013  |  0.294$\pm$0.018 |  0.379$\pm$0.020  |  0.450$\pm$0.024 |  0.555$\pm$0.026  |  0.718$\pm$0.029  |  0.834$\pm$0.022 |
> > > | $N=100$   | 0.186$\pm$0.011  |  0.321$\pm$0.019 |  0.396$\pm$0.023  |  0.493$\pm$0.023 |  0.602$\pm$0.027  |  0.759$\pm$0.027  |  0.872$\pm$0.021 |
> > > | $N=120$   | 0.185$\pm$0.010  |  0.331$\pm$0.017 |  0.426$\pm$0.019  |  0.501$\pm$0.022 |  0.626$\pm$0.023  |  0.793$\pm$0.017  |  0.901$\pm$0.011 |
> > > | $N=150$   | 0.200$\pm$0.010  |  0.330$\pm$0.012 |  0.424$\pm$0.015  |  0.520$\pm$0.016 |  0.641$\pm$0.018  |  0.807$\pm$0.016  |  0.907$\pm$0.011 |
> > >
> > > From the above results, we can see that the test power will increase in general when increasing $N$ from $20$ to $150$. The lowest test power appears when $N=20$ (0.459), and the highest test power appears when $N=150$ (0.907). This means that increasing the number of meta tasks will help improve the test power on the target task.

---

### Decision · Program_Chairs · 2021-09-27

**Decision:**

Accept (Poster)

**Comment:**

The main positive aspect of this paper is its inherent novelty: it is a new and unique way of improving two-sample testing by utilizing many related tasks. The corresponding theory also helps to demonstrate that the method is sound. Through thorough discussion with the reviewers, some additional experiments have been provided, and I ask that you please include these in the paper. The main lingering issue amongst the reviewers is the real-world motivation, or lack thereof. Some of the reviewers struggled to find a real-world scenario in which this kind of data would be present. During internal discussions, one reviewer did mention detecting adversarial examples, or changing distributions in streaming data as possible applications, which I thought worth mentioning here in case it is helpful.